# High-resolution kinetics of herbivore-induced plant volatile transfer reveal clocked response patterns in neighboring plants

Jamie Mitchel Waterman[1]*, Tristan Michael Cofer[1], Lei Wang[1], Gaetan Glauser[2], Matthias Erb[1]*

[1]Institute of Plant Sciences, University of Bern, Bern, Switzerland; [2]Neuchâtel Platform of Analytical Chemistry, Faculty of Science, University of Neuchâtel, Neuchâtel, Switzerland

*For correspondence: jamie.waterman@unibe.ch (JMW); matthias.erb@unibe.ch (ME)

**Competing interest:** The authors declare that no competing interests exist.

**Abstract** Volatiles emitted by herbivore-attacked plants (senders) can enhance defenses in neighboring plants (receivers), however, the temporal dynamics of this phenomenon remain poorly studied. Using a custom-built, high-throughput proton transfer reaction time-of-flight mass spectrometry (PTR-ToF-MS) system, we explored temporal patterns of volatile transfer and responses between herbivore-attacked and undamaged maize plants. We found that continuous exposure to natural blends of herbivore-induced volatiles results in clocked temporal response patterns in neighboring plants, characterized by an induced terpene burst at the onset of the second day of exposure. This delayed burst is not explained by terpene accumulation during the night, but coincides with delayed jasmonate accumulation in receiver plants. The delayed burst occurs independent of day:night light transitions and cannot be fully explained by sender volatile dynamics. Instead, it is the result of a stress memory from volatile exposure during the first day and secondary exposure to bioactive volatiles on the second day. Our study reveals that prolonged exposure to natural blends of stress-induced volatiles results in a response that integrates priming and direct induction into a distinct and predictable temporal response pattern. This provides an answer to the long-standing question of whether stress volatiles predominantly induce or prime plant defenses in neighboring plants, by revealing that they can do both in sequence.

## eLife assessment

This **fundamental** study examines the effects of herbivory-induced maize volatiles on neighbouring plants and their responses over time. Measurements of volatile compound classes and gene expression in receiver plants exposed to these volatiles led to the conclusion that the delayed emission of certain terpenes in receiver plants after the onset of light may be a result of stress memory, highlighting the role of priming and induction in plant defences triggered by herbivore-induced plant volatiles. The evidence supporting the conclusions is **compelling**, with rigorous chemical assays of and state-of-the-art high throughput real time mass spectrometry. The work will be of broad interest to plant biologists and chemical ecologists.

**eLife digest** Most plants are anchored to the soil by roots and need to be able to defend themselves from insects and other animal pests while remaining stationary. One way plants achieve this is to emit chemicals known as herbivore-induced plant volatiles (HIPVs) into the air when they are under attack to attract other animals that are natural enemies of the pest.

Certain HIPVs also prime other nearby plants (known as 'receivers') to be ready for an attack, or even pre-emptively activate defense responses in the plant before they encounter the pest. However, it remains unclear how the temporal patterns of HIPVs emitted from attacked plants affect how receiver plants respond to these chemicals, and how day-to-night light fluctuations impact this transfer of chemical information.

To investigate this question, Waterman et al. exposed maize plants to a common pest caterpillar called *Spodoptera exigua*. Individual infested maize plants (referred to as 'senders') were placed in transparent glass chambers that were linked by a narrow tube to a second glass chamber containing a receiver plant that had not encountered caterpillars. The team used a mass spectrometry approach to measure the HIPVs emitted by the sender plants and the responses of the receivers in real-time.

The experiments found that within the first few hours of exposure to HIPVs, receiver plants had a small burst of defense activity that was followed by a far stronger burst several hours later. The second burst coincided with the accumulation of plant hormones called jasmonates in the receiver plants, and was not controlled by fluctuations in light levels. This suggests that HIPVs first prime and then subsequently induce defense responses in other plants in a manner that is independent of the patterns of day and night.

In the future, these findings may be used to aid in the diagnosis and monitoring of pest outbreaks in crop fields. They will also help us to better understand how plants communicate and the impact of this communication on their environment.

## Introduction

Plants rely on chemical cues to identify and mount appropriate responses to herbivores (*Waterman et al., 2019*; *Escobar-Bravo et al., 2023*). Among these cues are herbivore-induced plant volatiles (HIPVs) (*Meents and Mithöfer, 2020*; *Karban, 2021*). When perceived by undamaged plants, HIPVs can enhance plant defenses, and thus increase plant resistance to future herbivory (*Erb et al., 2015*; *Kalske et al., 2019*; *Hu et al., 2019*; *Engelberth et al., 2004*; *Vázquez-González et al., 2023*). HIPVs can directly induce plant defenses (*Hu et al., 2019*; *Engelberth et al., 2004*; *Wang et al., 2023*; *Ruther and Fürstenau, 2005*). HIPVs can also prime plant defenses. In this case, HIPV-exposed plants respond more strongly and/or more rapidly to a secondary stimulus such as herbivore attack (*Erb et al., 2015*; *Hu et al., 2019*; *Engelberth et al., 2004*; *Mauch-Mani et al., 2017*).

Plant responses to environmental cues depend on temporal patterns. For example, artificial herbivory that mimics spatiotemporal patterns of chewing herbivores induces more a similar response to actual herbivore attack than when spatiotemporal patterns are ignored (*Mithöfer et al., 2005*). The emission of HIPVs also follows discrete temporal dynamics (*Erb et al., 2015*; *Yu et al., 2017*), which, again, is likely to influence defense responses in receiver plants. Green leaf volatiles (GLVs), including (*Z*)-3-hexenyl acetate (HAC), (*Z*)-3-hexenal, and (*Z*)-3-hexen-1-ol, are catabolic products of the lipoxygenase/hydroperoxide lyase pathway and are emitted within minutes following damage (*D'Auria et al., 2007*; *Matsui, 2006*). Indole and terpenes are emitted within hours of herbivore attack (*Escobar-Bravo et al., 2023*; *Erb et al., 2015*). Indole can mediate interactions between plants as well as between plants and other organisms (*Ye et al., 2021*; *Ye et al., 2018*; *Veyrat et al., 2016*). Terpenes are a very diverse group of volatiles (>80,000 known structures) that are also important as both direct and indirect plant defenses (*Kalske et al., 2019*; *Chen et al., 2021*; *Zhou and Pichersky, 2020*; *Riedlmeier et al., 2017*). Indole and terpenes take longer to produce in comparison to GLVs and are less transiently emitted (*Escobar-Bravo et al., 2023*; *Erb et al., 2015*; *D'Auria et al., 2007*). In maize, GLVs have been implicated in direct induction (*Engelberth et al., 2004*; *Wang et al., 2023*), and both GLVs and indole are known to prime defenses, and are thus considered as 'bioactive' in information transfer between plants (*Erb et al., 2015*; *Hu et al., 2019*).

Although it is clear that plants can either be primed or directly induced following HIPV exposure, if and how these two phenomena integrate in the context of a natural HIPV exposure sequence is unclear. Studies have yet to monitor induction and priming dynamics in real time and during continuous exposure to HIPVs. It is possible that, through time, continuous HIPV exposure may result in a self-enforced positive feedback loop, whereby priming enhances induction and induction serves as a priming event for subsequent induction. Further, temporal trends might be complicated by dynamic environmental conditions, such as light fluctuations; over a 24 hr period, plants can experience both full sun and darkness. In darkness, the emission of many volatiles, especially those released via stomata, is severely hindered (*Seidl-Adams et al., 2015*; *He et al., 2021*; *Bläsing et al., 2005*; *Lin et al., 2022*).

In order to understand the kinetics of information transfer between plants and the importance of dynamic temporal emission patterns, we monitored the emission kinetics of terpenes as a marker of defense activation in maize plants that were exposed to HIPVs. We leveraged a highly temporally resolved volatile multiplexing system that allowed us to track continuous volatile emissions in sender and receiver plants for the first time in real time. We further measured foliar terpene pools, stress hormones, and defense gene expression in receiver plants, and determined the impact of day:night light transitions in the observed defense induction kinetics. Finally, we measured defenses upon sequences of short-term HIPV and GLV exposure to unravel the relative importance of direct induction and priming. Together, our experiments reveal how the natural kinetics of herbivore attack result in a clocked defense response in neighboring plants via volatile information transfer.

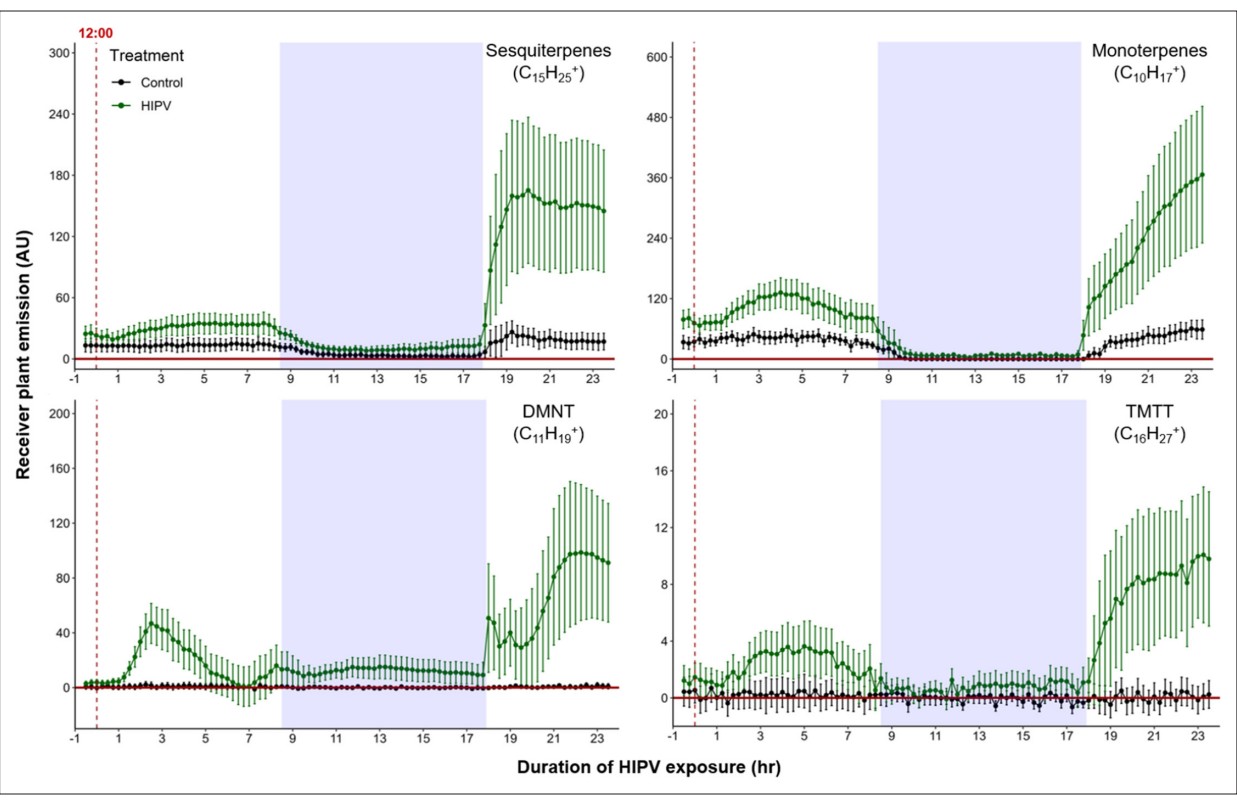

**Figure 1.** Delayed burst in induced volatile emissions in plants exposed to volatiles of a herbivore-attacked neighbor. Emission kinetics of herbivore-induced plant volatile (HIPV)-induced terpenes in undamaged receiver plants are shown. Dark green points represent mean emission of herbivore-damaged sender plants connected to undamaged receiver plants, with the emissions from damaged sender plants only subtracted. Black points represent undamaged sender plants connected to undamaged receiver plants, with the emissions from undamaged sender plants only subtracted. Blue rectangles represent the night (dark phase). Abbreviations: DMNT, 4,8-dimethylnona-1,3,7-triene; TMTT, 4,8,12-trimethyltrideca-1,3,7,11-tetraene. Error bars = SE. n=8–10. Compounds were identified based on their molecular weight+1, as all compounds were protonated. Sesquiterpenes: m/z=205.20; monoterpenes: m/z=137.13; DMNT: m/z=151.15; TMTT: m/z=219.21.

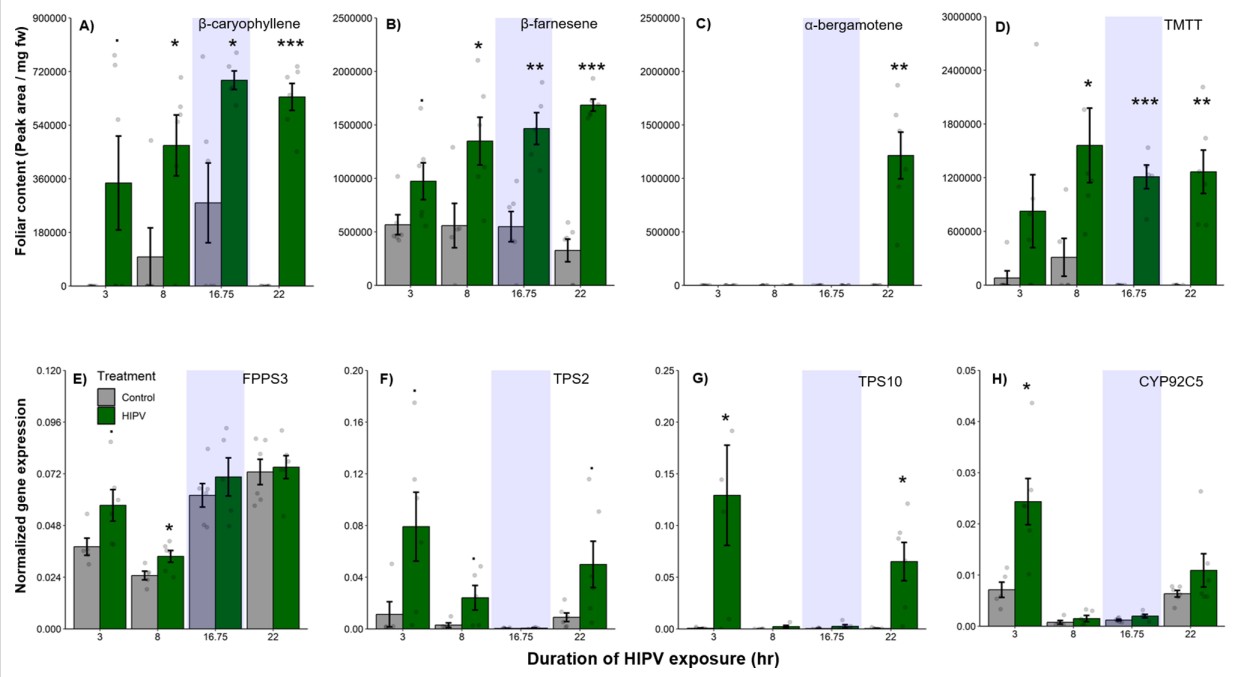

**Figure 2.** The delayed burst in terpene emission is not explained by terpene overaccumulation during the night. Accumulation of terpenes and induction of terpene biosynthesis genes in receiver plants was measured over time. (**A–D**) Internal foliar pools of terpenes in receiver plants. (**E–H**) Expression of terpene biosynthesis genes in receiver plants. Dark green bars represent receiver plants exposed to herbivore-induced plant volatiles (HIPVs) and light gray bars represent receiver plants connected to undamaged sender plants. Blue rectangles represent the night (dark phase). Abbreviations: TMTT, 4,8,12-trimethyltrideca-1,3,7,11-tetraene; FPPS3, farnesene pyrophosphate synthase 3; TPS2, terpene synthase 2; TPS10, terpene synthase 10; CYP92C5, dimethylnonatriene/trimethyltetradecatetraene synthase. ˙ = p<0.1, *=p<0.05, **=p<0.01, ***=p<0.001 as determined by a Welch's two-sample *t*-test. Bars = mean ± SE. n=4–6.

## Results

### Delayed burst in induced volatile emissions in plants exposed to volatiles of a herbivore-attacked neighbor

When maize plants are exposed to volatiles from herbivore-attacked neighbors, they start to release terpenes (*Hu et al., 2019*; *Ruther and Fürstenau, 2005*). To understand the temporal dynamics of this phenomenon, we conducted a detailed time-course analysis of terpene release by receiver plants exposed to volatiles from sender plants under attack by *Spodoptera exigua* caterpillars (*Figure 1*). Herbivore feeding began around 12:00 and within 3 hr, we detected a small induction of sesquiterpenes [$C_{15}H_{25}^+$, m/z=205.20], monoterpenes [$C_{10}H_{17}^+$, m/z=137.13], 4,8-dimethylnona-1,3,7-triene (DMNT) [$C_{11}H_{19}^+$, m/z=151.15], and 4,8,12-trimethyltrideca-1,3,7,11-tetraene (TMTT) [$C_{16}H_{27}^+$, m/z=219.21] in undamaged receiver plants. Volatile emission was severely impaired during the night, regardless of treatment. Interestingly, as soon as the light was restored, we observed a strong burst of terpene release from receiver plants, which was two to five times higher than the release the day before (*Figure 1*). Thus, exposure to volatiles from a herbivore-attacked plant triggers a delayed burst in terpene emission in neighboring plants.

### The delayed burst in terpene emission is not explained by overaccumulation during the night

Why do receiver plants show a delayed burst in volatile-induced terpenoid release? During the night, plants close their stomata to limit water loss (*Caird et al., 2007*), which may also impair terpene release, thus leading to an accumulation of terpenes in the leaves and a burst once stomata open again (*Seidl-Adams et al., 2015*; *Loughrin et al., 1994*). To determine if terpenes accumulate above daytime levels in maize leaves of HIPV-exposed plants during the night, we measured internal foliar pools of the sesquiterpenes, β-caryophyllene, β-farnesene, and α-bergamotene, as well as the homoterpene,

**Table 1.** Welch's two-sample *t*-test results comparing foliar terpene pools, biosynthesis genes, and phytohormone levels between herbivore-induced plant volatile (HIPV)-exposed and control receiver plants.

Bold values: p<0.05, underlined values: p<0.1. Abbreviations: β-car, β-caryophyllene; β-farn, β-farnesene; α-berg, α-bergamotene; TMTT, 4,8,12-trimethyltrideca-1,3,7,11-tetraene; FPPS3, farnesene pyrophosphate synthase 3; TPS2, terpene synthase 2; TPS10, terpene synthase 10; CYP92C5, dimethylnonatriene/trimethyltetradecatetraene synthase; OPDA, 12-oxophytodienoic acid; JA, jasmonic acid; JA-Ile, jasmonic acid-isoleucine; OPR7, oxo-phytodienoate reductase 7.

| | Time | | | | | | | | | | | |
|---|---|---|---|---|---|---|---|---|---|---|---|---|
| Response | 3 hr | | | 8 hr | | | 16.75 hr | | | 22 hr | | |
| | t | p | df | t | p | df | t | p | df | t | p | df |
| β-car | −2.20 | <u>0.08</u> | 5 | −2.65 | **0.03** | 9 | −3.00 | **0.03** | 6 | −14.05 | **<0.001** | 5 |
| β-farn | −2.08 | <u>0.07</u> | 8 | −2.60 | **0.03** | 9 | −4.47 | **0.002** | 9 | −11.38 | **<0.001** | 8 |
| α-berg | - | - | - | - | - | - | - | - | - | −5.56 | **0.003** | 5 |
| TMTT | −1.80 | 0.13 | 5 | −2.68 | **0.03** | 7 | −9.18 | **<0.001** | 4 | −5.23 | **0.003** | 5 |
| FPPS3 | −2.31 | <u>0.05</u> | 8 | −2.62 | **0.03** | 9 | −0.82 | 0.44 | 7 | −0.28 | 0.78 | 10 |
| TPS2 | −2.39 | <u>0.05</u> | 6 | −2.19 | <u>0.09</u> | 4 | −0.62 | 0.55 | 8 | −2.24 | <u>0.07</u> | 5 |
| TPS10 | −2.65 | **0.05** | 5 | −1.97 | 0.12 | 4 | −1.34 | 0.25 | 4 | −3.50 | **0.02** | 5 |
| CYP92C5 | −3.62 | **0.01** | 6 | −1.11 | 0.30 | 8 | −1.94 | 0.11 | 5 | −1.38 | 0.22 | 5 |
| OPDA | −0.26 | 0.80 | 4 | 0.07 | 0.95 | 5 | 0.65 | 0.55 | 4 | −4.18 | **0.008** | 5 |
| JA | 0.32 | 0.78 | 2 | 1.22 | 0.31 | 3 | 0.91 | 0.41 | 4 | −1.40 | 0.22 | 5 |
| JA-Ile | −0.48 | 0.68 | 2 | 1.21 | 0.31 | 3 | 0.24 | 0.82 | 5 | −2.98 | **0.04** | 4 |
| OPR7 | −5.70 | **0.002** | 5 | −2.36 | <u>0.05</u> | 7 | 1.32 | 0.22 | 10 | −3.01 | **0.02** | 7 |

TMTT, over time. Internal pools of β-caryophyllene, β-farnesene, and TMTT were marginally induced after 3 hr, but only became significantly induced after 8 hr of exposure to HIPVs (*Figure 2A, B, and D*, *Table 1*). For all three, accumulation remained higher during the night (16.75 hr) as well as the following day. α-Bergamotene only began accumulating in leaves on the second day (*Figure 2C*, *Table 1*). Thus, internal terpene pools remain comparable between night and day, suggesting that terpenes do not continue accumulating during the night and that, even when terpenes are emitted in large amounts on day 2, internal pools do not drop below nighttime levels. Nevertheless, the volatiles that are present during the night may be released suddenly the second day. Some control plants accumulated measurable amounts of β-caryophyllene at night, likely due to biological and experimental variation. This type of variation was not visible in total sesquiterpene emissions measured by proton transfer reaction time-of-flight mass spectrometry (PTR-ToF-MS) (*Figure 1*), likely because β-caryophyllene is a minor sesquiterpene in B73 (*Block et al., 2018*) and closed stomata block sesquiterpene emission at night (*Seidl-Adams et al., 2015*). To get insight into terpene biosynthesis we measured the expression of terpene synthases in receiver plants. In maize, farnesene pyrophosphate synthase 3 (FPPS3), terpene synthase 2 and 10 (TPS2 and TPS10, respectively), as well as dimethylnonatriene/trimethyltetradecatetraene synthase (CYP92C5) are rate limiting for terpene production (*Block et al., 2019*; *Richter et al., 2015*; *Richter et al., 2016*). All genes were induced by HIPV exposure during daytime, but not during the night. FPPS3 was slightly induced after 3 hr and significantly induced after 8 hr of HIPV exposure (*Figure 2E*, *Table 1*). TPS2 was slightly induced 3, 8, and 22 hr after the onset of HIPV exposure (*Figure 2F*, *Table 1*). TPS10 was significantly induced after 3 and 22 hr of HIPV exposure (*Figure 2G*, *Table 1*). CYP92C5 was only significantly induced after 3 hr of HIPV exposure (*Figure 2H*, *Table 1*). The patterns of volatile terpene biosynthesis do not support a scenario where the terpene burst on the second day is due to continued biosynthesis but lack of emission during the night.

## The delayed burst in terpenoid emission is associated with clocked jasmonate production

Volatile release in maize is regulated by jasmonates (*Martin et al., 2003*; *Ament et al., 2004*). To understand whether the delayed terpene burst is associated with jasmonate signaling, we measured the levels of 12-oxo-phytodienoic acid (OPDA), jasmonic acid (JA), and jasmonic acid-isoleucine (JA-Ile), as well as the expression of oxo-phytodienoate reductase 7 (OPR7), which is critical for JA

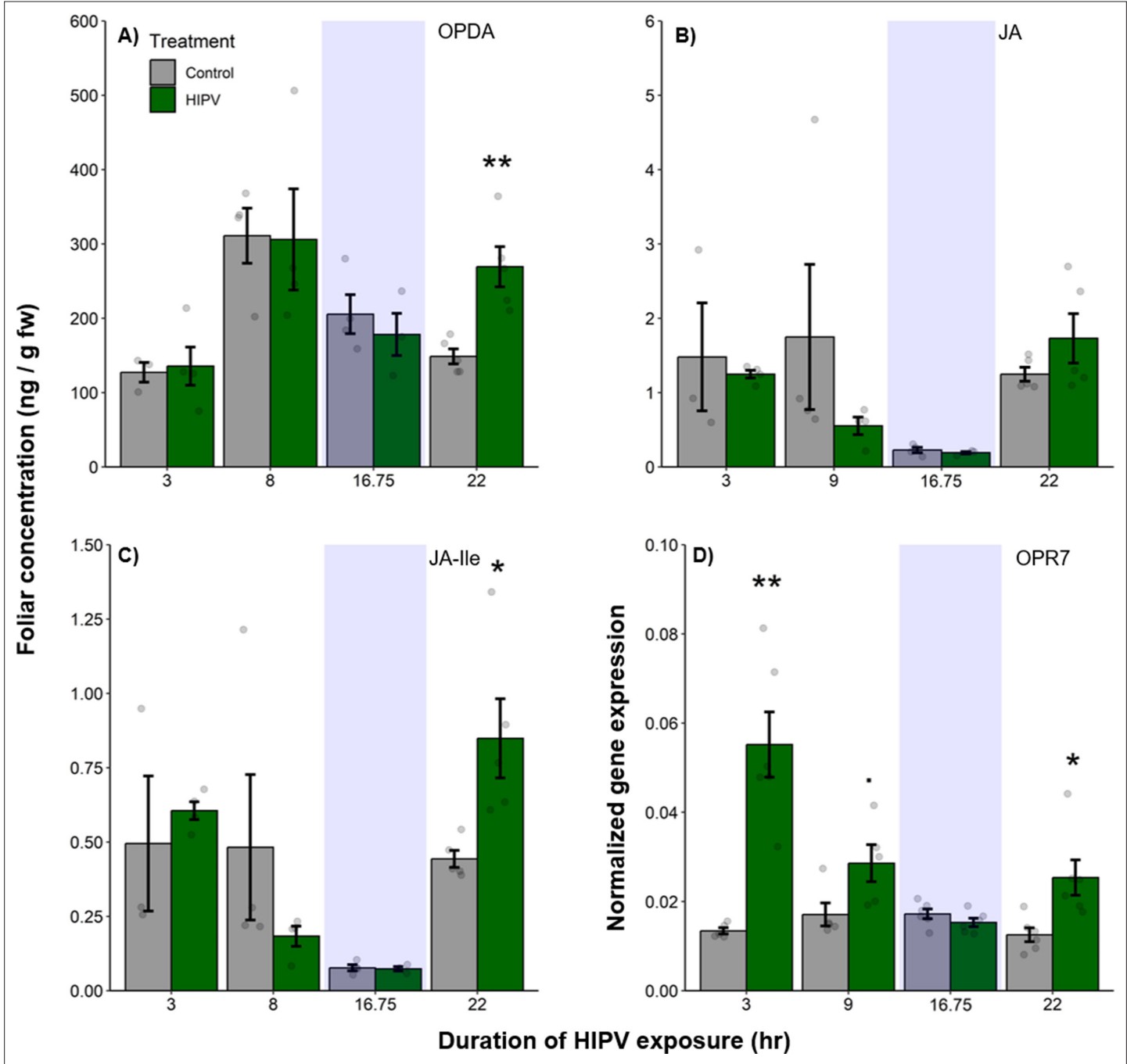

**Figure 3.** The delayed burst in terpene emission is associated with clocked jasmonate production. Foliar jasmonate concentrations (**A–C**) and jasmonate biosynthesis (OPR7; **D**) in receiver plants over time are shown. Dark green bars represent receiver plants exposed to herbivore-induced plant volatiles (HIPVs) and light gray bars represent receiver plants connected to undamaged sender plants. Blue rectangles represent the night (dark phase). Abbreviations: OPDA, 12-oxophytodienoic acid; JA, jasmonic acid; JA-Ile, jasmonic acid-isoleucine; OPR7, oxo-phytodienoate reductase 7. = p<0.1, *=p<0.05, **=p<0.01, ***=p<0.001 as determined by a Welch's two-sample *t*-test. Bars = mean ± SE. n=3–6.

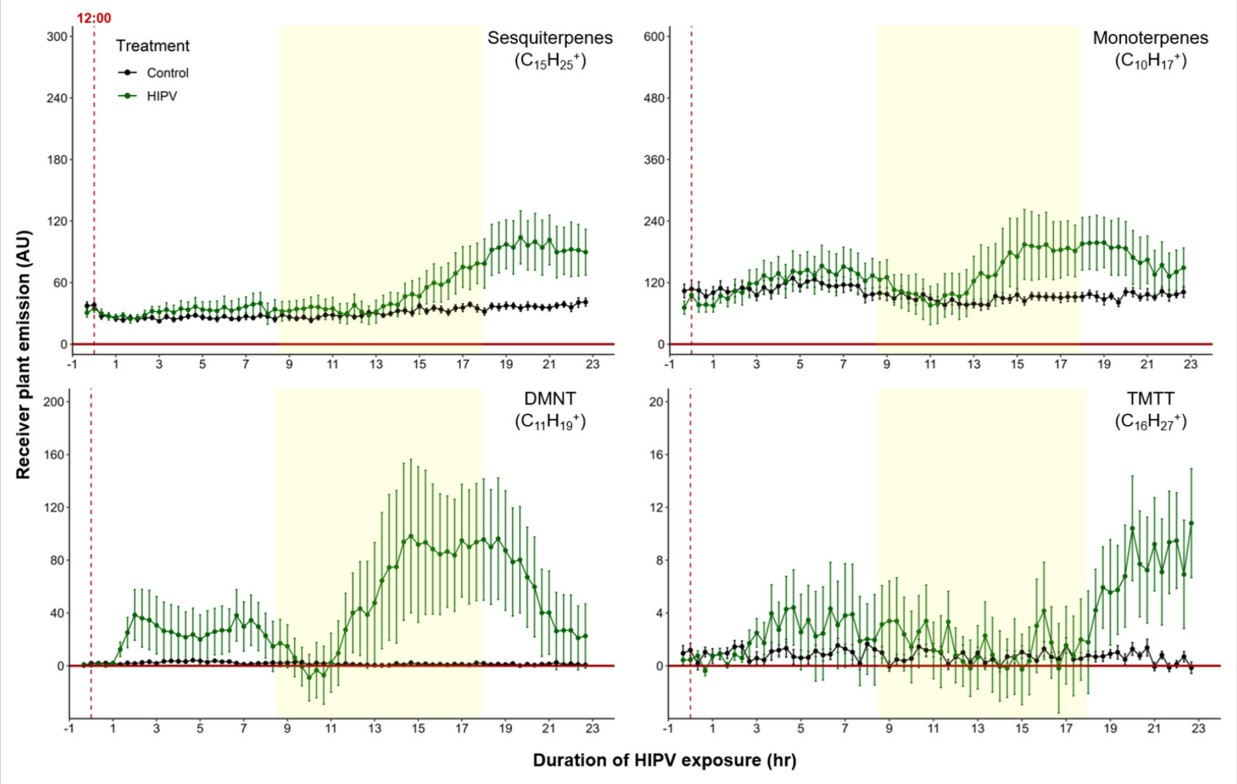

**Figure 4.** The delayed volatile burst is conserved under continuous light. Emission kinetics of herbivore-induced plant volatile (HIPV)-induced terpenes in undamaged receiver plants under continuous light are shown. Plants were grown under normal light conditions. Lights were left on continuously following the start of the treatment. Dark green points represent the mean emission of herbivore damaged sender plants connected to undamaged receiver plants with the emissions from damaged sender plants only subtracted. Black points represent undamaged sender plants connected to undamaged receiver plants, with the emissions from undamaged sender plants only subtracted. Yellow rectangles represent when the lights would typically be turned off. Abbreviations: DMNT, 4,8-dimethylnona-1,3,7-triene; TMTT, 4,8,12-trimethyltrideca-1,3,7,11-tetraene. Error bars = SE. n=8–10. Compounds were identified based on their molecular weight+1, as all compounds were protonated. Sesquiterpenes: m/z=205.20; monoterpenes: m/z=137.13; DMNT: m/z=151.15; TMTT: m/z=219.21.

The online version of this article includes the following figure supplement(s) for figure 4:

**Figure supplement 1.** The delayed volatile burst is conserved under continuous light, regardless of the time of day.

biosynthesis (*Yan et al., 2012*). We found a significant induction of OPDA and JA-Ile production as well as OPR7 expression at the beginning of the second day, 22 hr after the onset of volatile exposure (*Figure 3*, *Table 1*). Thus, jasmonate production is temporally aligned with the delayed terpene burst at the onset of the second day.

## The delayed volatile burst is conserved under continuous light

To test whether the delayed burst in terpene volatiles is linked to light:dark transitions, we exposed maize plants to HIPVs under continuous light. Similar to exposure during a normal light regime, the largest burst of terpene emission occurred *ca.* 12–18 hr after the onset of volatile exposure (*Figure 4*), suggesting that the temporal delay in volatile emission in receiver plants is not dependent on light fluctuations, but is otherwise clocked. In order to control for potential differences in induction capacity at different times of day, we designed a second continuous light experiment. However, instead of starting the herbivory treatment at ca. 12:00 hr, we started it around 20:00 hr. Similarly, we observed an initial bump in terpene emission followed by a larger burst 10–12 hr after HIPV exposure (*Figure 4— figure supplement 1*). In both continuous light experiments the 'second day' burst occurred somewhat earlier than under normal light conditions. It is possible that stomata, which are closed in the dark, delay the onset of the terpene burst under dark:light transitions.

## The delayed volatile burst cannot be fully explained by the emission kinetics of bioactive herbivory-induced volatiles

A simple explanation for the delayed volatile burst may be that the bioactive volatiles are more strongly emitted from sender plants at the onset of day 2, thus triggering a stronger volatile response in the receiver plants at this time. To test this hypothesis, we analyzed volatile emission kinetics of *S. exigua*-infested plants over time. We focused our analysis on GLVs and indole, which are known to induce and/or prime volatile release in neighboring maize plants (*Hu et al., 2019*; *Engelberth et al., 2004*; *Wang et al., 2023*; *Ruther and Fürstenau, 2005*). Terpenes are not known to prime or induce volatile release in maize, and were thus not included in the sender plant analysis (*Ruther and Fürstenau, 2005*). We analyzed GLV and indole emission data of the sender plants from the experiments depicted in *Figure 1* and *Figure 4*, *Figure 5*, and then correlated their emission with the induction of terpenes in receiver plants (*Figure 6*). We found strong positive correlations between hexenyl acetate, hexenal, hexen-1-ol, and indole emissions in sender plants and terpene responses in receiver plants when plants had a dark period (*Figure 6*). Interestingly, this was not the case under continuous light, where the apex of bioactive volatile emission in sender plants was not temporally aligned with the apex of terpene responses in receiver plants (*Figure 6*). Sender plant emissions even showed slight negative correlations with terpene induction in receiver plants under continuous light. Thus, the emission kinetics of bioactive volatiles from sender plants cannot fully explain the delayed terpene burst.

## The combination of direct induction and priming can explain the delayed terpene burst in receiver plants

Based on the above observations, we reasoned that, under natural conditions, the strong delayed volatile burst may be due to the interaction of an initial volatile burst that primes plants for higher volatile release at the onset of the second day, and a secondary trigger in the form of bioactive volatiles released from sender plants at the onset of the second day. To test this hypothesis, we exposed receiver plants to volatiles from control plants or volatiles from herbivory-induced plants for 1.25 hr. We then disconnected the receiver plants and exposed them to clean air for 17 hr, until the beginning of the next day. Half of the plants were then exposed to HAC as a secondary trigger. HAC was selected as an inducer as it showed the strongest correlation with all measured terpenes (*Figure 6*). Following the short exposure to HIPVs, we detected a generally small, but significant, induction of sesquiterpenes, monoterpenes, DMNT, and TMTT emissions in receiver plants (*Figure 7*, *Table 2*). During the night, no differences between treatments were detected any more. At the beginning of the next day, we found a slight induction in terpene emissions, most apparent for sesquiterpenes and TMTT, in plants that had been exposed to herbivory-induced volatiles 18 hr prior (*Figure 7*, *Table 2*). HAC exposure at the beginning of the next day also induced terpene emissions. Furthermore, HAC induced a stronger release of all terpenes in plants that had been exposed to HIPVs the previous day. In an independent experiment, we also tested whether HAC sensitivity per se may be higher at the start of the light period. We exposed maize seedlings to HAC dispensers either 30 min or 4 hr 30 min after lights came on. We found induction in terpene emission to be similar regardless of when induction began (*Figure 7—figure supplement 1*). Thus, exposure to HIPVs prompts maize plants to release more terpenes the next day and also primes maize plants to respond more strongly to a secondary HIPV trigger. Together, these two phenomena result in a pronounced terpene burst.

## Discussion

HIPVs play an important role in mediating interactions between damaged and undamaged plants (*Escobar-Bravo et al., 2023*). However, the kinetics and temporal dynamics of this information transfer remain poorly understood. We show that, somewhat surprisingly, upon continuous exposure to HIPVs, receiver plants show a pronounced activation of defenses at the onset of the second day. We find that this is the result of volatile-mediated priming on the first day that yields a clocked response to the volatiles that are perceived the next day. Here, we discuss the mechanisms and biological implications of this phenomenon.

Plants are well known to respond to herbivore-induced volatiles by increasing their own defenses. Defense activation can happen directly, with plants increasing their defense hormone production and

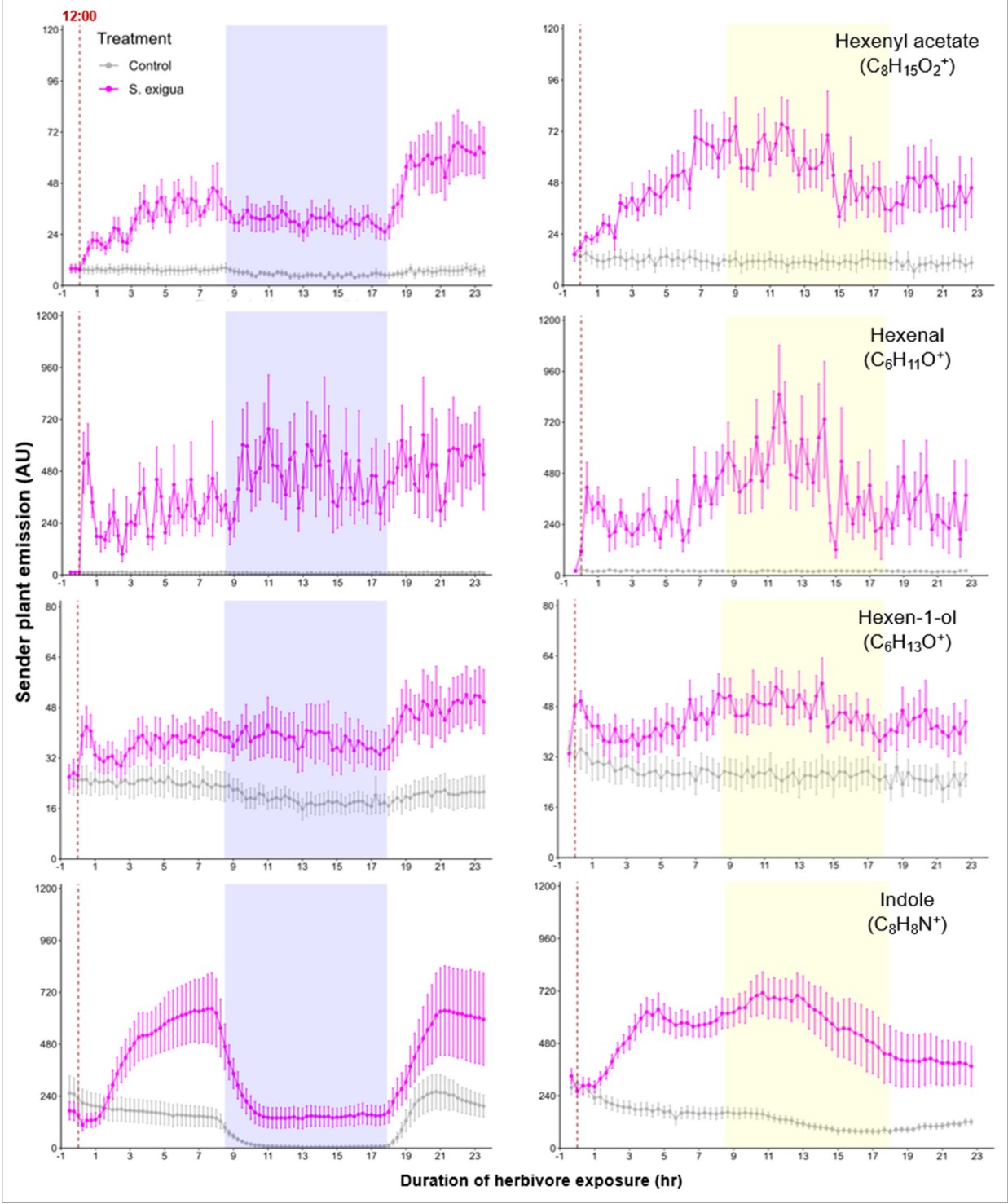

**Figure 5.** Green leaf volatile (GLV) and indole emissions in *S. exigua*-damaged plants. Pink points represent the mean emission of herbivore-damaged sender plants. Gray points represent mean emissions of undamaged sender plants. Blue rectangles represent the night (dark phase). Yellow rectangles represent when the lights would typically be turned off. Continuous light-exposed plants were grown under normal light conditions, however lights were left on continuously following the start of the treatment. Error bars = SE. n=8–10. Compounds were identified based on their molecular weight+1, as all compounds were protonated. Sesquiterpenes: m/z=205.20; monoterpenes: m/z=137.13; DMNT: m/z=151.15; TMTT: m/z=219.21.

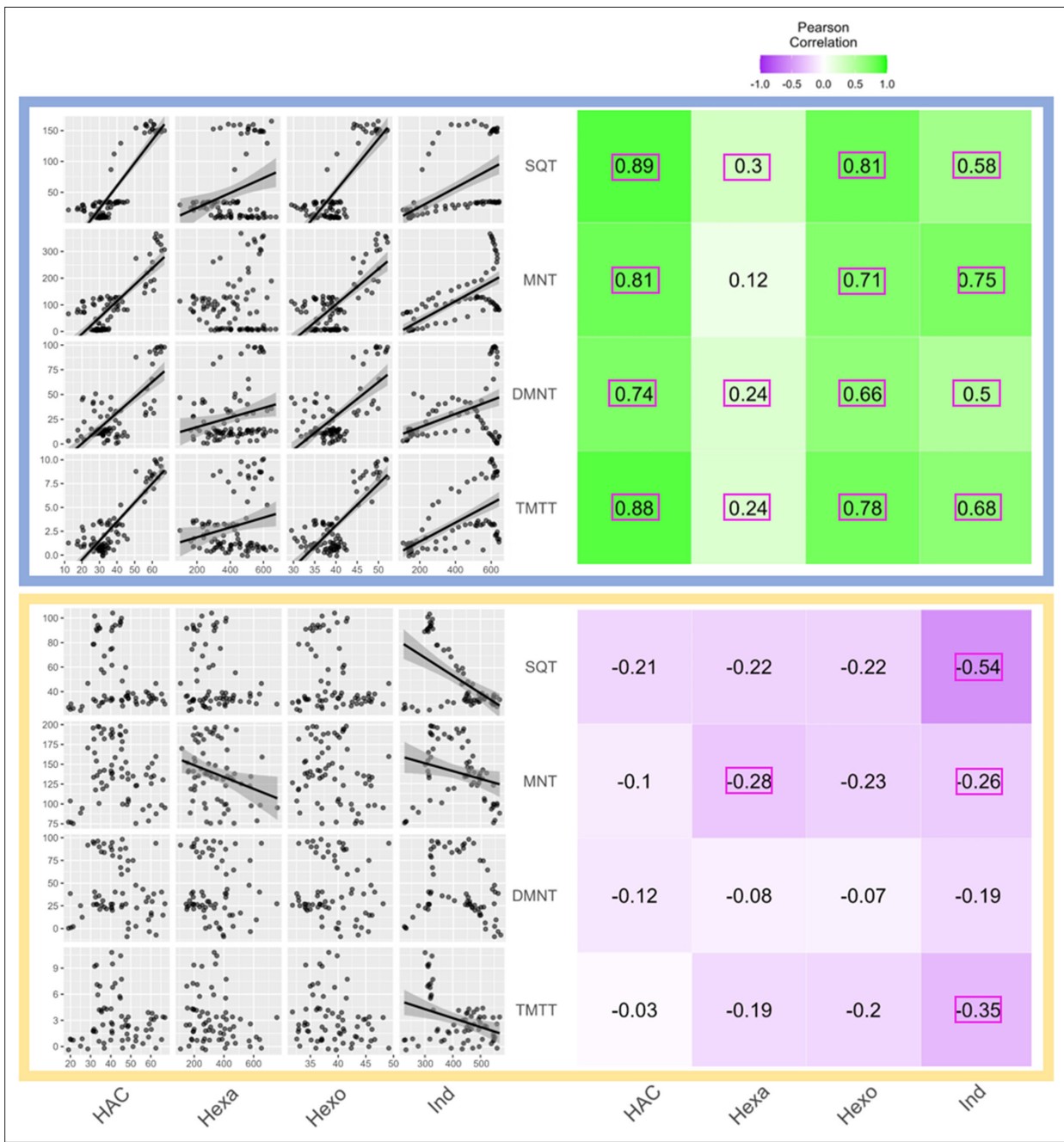

**Figure 6.** Receiver plant terpene emissions are tightly correlated with bioactive sender plant signals under normal light conditions. The left panels depict scatter plot correlation matrices of bioactive volatile emissions from damaged sender plants and terpene emissions from herbivore-induced plant volatile (HIPV)-exposed receiver plants. Only data from the first measurement following the addition of herbivores to sender plants are included. Upper scatter plot (blue box) shows correlations under normal light conditions and lower scatter plot (yellow box) depicts correlations under continuous light. For continuous light-exposed plants, lights were left on continuously following the start of the treatment. Each black point represents the mean value of all individuals at a given time point after herbivory began. Regression curves are only shown for significant relationships (p<0.05). Panels on the right-hand side depict heat maps based on the value of Pearson's correlation coefficient between two given compounds. Numbers in the center of each square are Pearson's correlation coefficient. Correlation coefficients contained in a pink rectangles indicate a significant correlation (p<0.05). (Z)-3-hexenyl acetate (HAC), hexenal (Hexa), hexen-1-ol (Hexo), and indole (Ind) were from sender plants, and sesquiterpenes (SQT), monoterpenes (MNT), 4,8-dimethylnona-1,3,7-triene (DMNT), and 4,8,12-trimethyltrideca-1,3,7,11-tetraene (TMTT) were from receiver plants.

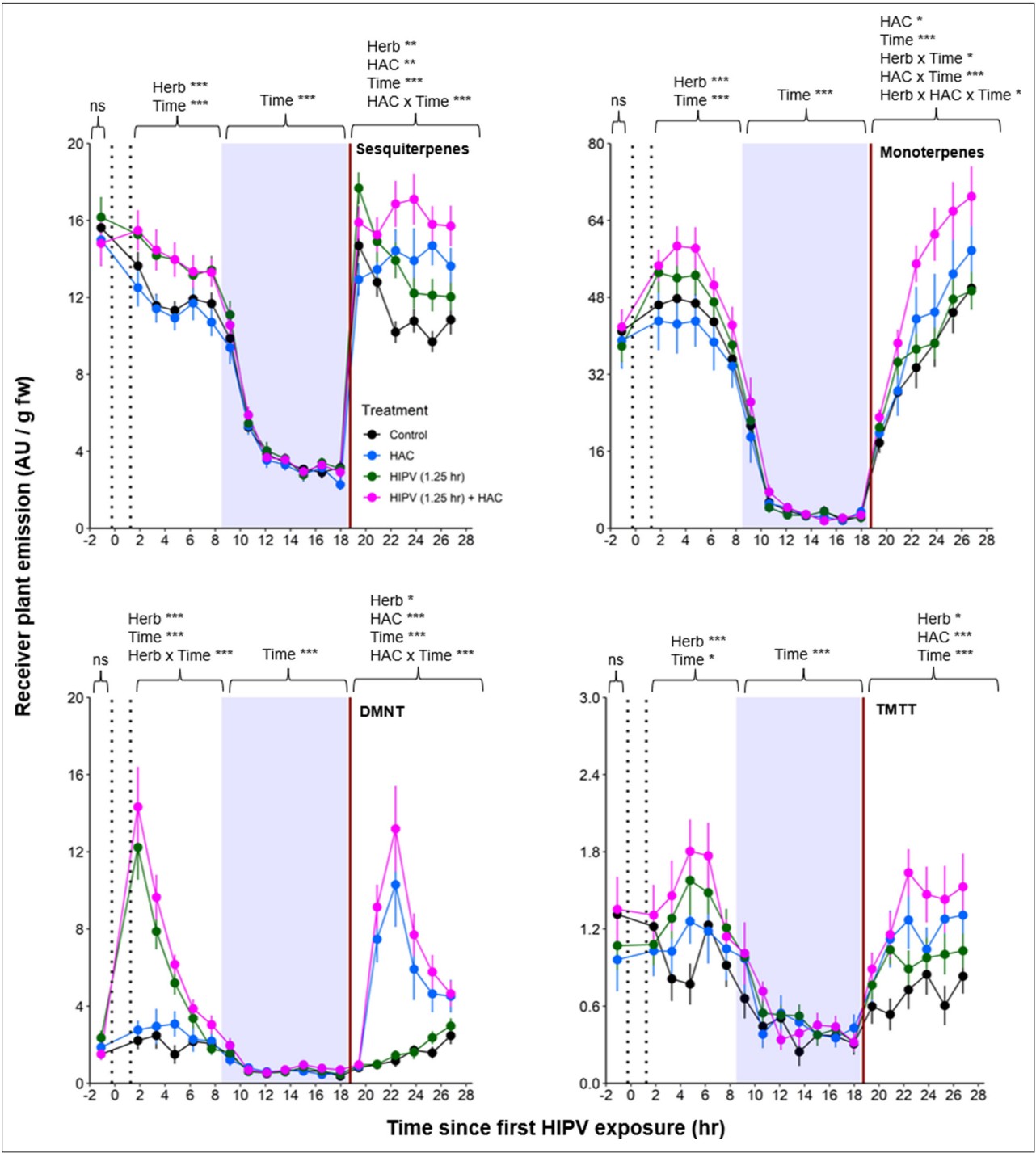

**Figure 7.** The combination of volatile priming and sender emission kinetics can explain the delayed terpene burst in receiver plants. Sender plants were connected to receiver plants 30 min prior to herbivore exposure on sender plants and left connected for 1.25 hr following exposure (time between perforated vertical lines). After 1.25 hr, chambers were disconnected and measurements were collected from receiver plant chambers only. The following day, after light was restored, plants were treated with (Z)-3-hexenyl acetate (HAC) dispensers to simulate bioactive signals (indicated by red solid vertical line). Blue rectangles represent the night (dark phase). *=p<0.05, **=p<0.01, ***=p<0.001 as determined by aligned rank transformed nonparametric factorial repeated measures ANOVA. Abbreviations: HIPV, herbivore-induced plant volatile; DMNT, 4,8-dimethylnona-1,3,7-triene; TMTT, 4,8,12-trimethyltrideca-1,3,7,11-tetraene. Colored points represent mean emissions standardized by fresh weight (fw). Error bars = SE. n=12–16. Compounds were identified based on their molecular weight+1, as all compounds were protonated. Sesquiterpenes: m/z=205.20; monoterpenes: m/z=137.13; DMNT: m/z=151.15; TMTT: m/z=219.21.

The online version of this article includes the following figure supplement(s) for figure 7:

*Figure 7 continued on next page*

*Figure 7 continued*

**Figure supplement 1.** Maize seedlings respond consistently to green leaf volatiles (GLVs) throughout the day.

**Figure supplement 2.** Green leaf volatile (GLV) emission from herbivore-damaged maize seedlings is highly variable over time.

volatile release (*Engelberth et al., 2004*; *Wang et al., 2023*), and/or via priming effects, with plants increasing their defenses more strongly upon a secondary stimulus (*Erb et al., 2015*; *Hu et al., 2019*). Here, we show that these two phenomena operate together to trigger strong defense activation in receiver plants with predictable temporal kinetics. So far, studies in maize and other plants showing priming effects have often exposed receiver plants to HIPVs for several hours, overnight, or even for multiple days, and induced them, e.g., by simulated herbivory, on the following day in the absence of HIPVs (*Erb et al., 2015*; *Engelberth et al., 2004*; *Ton et al., 2007*; *Ali et al., 2013*; *Paudel Timilsena et al., 2020*). Indeed, such a setup would reveal clear priming, with minor or no direct induction by the volatile treatment. Our findings demonstrate that more natural continuous exposure to volatiles results in the same pattern without the need for another stimulus. We thus conclude that HIPVs are sufficient to trigger robust, clocked defense activation in neighboring plants.

What is the mechanism that results in a strong activation of defenses on the onset of the second day of exposure to HIPVs? In maize, GLVs directly induce and prime jasmonate production and terpene release (*Hu et al., 2019*; *Engelberth et al., 2004*; *Wang et al., 2023*). Indole can prime, but not directly induce these two responses (*Erb et al., 2015*). GLVs and indole again can interact to increase defense priming upon a secondary stimulus (*Hu et al., 2019*). Given these considerations, together with the observed volatile release, response kinetics, and our manipulative experiments, we can draw up the following scenario: On day 1, maize plants are exposed to GLVs, which trigger a small burst of terpene release in receiver plants. At the same time, receiver plants are primed to respond more strongly on the next day, once light is restored. At the onset of the second day, four things happen simultaneously. First, the emission of GLVs from the neighboring plants increases. Second, the defense priming mechanism responding to these cues kicks into gear. Third, terpene production is activated. Fourth, stomata open and enable volatile emission. Together, these elements result in a strong terpene burst. The orchestration of these elements is noteworthy and results in a predictable response pattern under variable HIPV exposure. Further experiments will reveal whether similar patterns are observed in other plants, and how they may operate on a mechanistic level. Additionally, considering we observed an induction in jasmonates coinciding with terpene bursts it is likely that non-volatile jasmonate-dependent defenses follow similar temporal patterns. Our experiments reveal the importance of tracking plant defense responses in real time as they perceive the dynamic natural volatile blend of herbivore-attacked neighbors.

What is the ecological relevance of the clocked temporal kinetics of defense activation in neighboring plants? We propose several hypotheses. First, responding most strongly to volatile cues on the second day may avoid unnecessary energy expenditure (*Waterman et al., 2019*; *Mithöfer et al., 2005*). GLVs are emitted upon both herbivore attack and mechanical damage (*Gardiner et al., 2016*). However, when damage is not sustained, GLVs will dissipate rapidly (*D'Auria et al., 2007*). Thus, responding most strongly to repeated GLV exposure could avoid false negatives and allow plants to respond more robustly to the presence of actual herbivory. Second, sending out volatiles on the second day may maximize indirect defenses. Attracting natural enemies too early to herbivore-free plants could have fitness costs for both the plant and herbivore natural enemies. Responding on the second day could be advantageous, as by this time, the natural enemies would have located and interacted with the herbivores on sender plants, and would thus be ready to move over to the next plant in anticipation of herbivore arrival. As fitness outcomes are likely highly nuanced, especially considering HIPVs also attract herbivores (*Zu et al., 2020*), investigation into information transfer on the multi-plant scale, namely in the context of multi-trophic interactions, will be critical. Considerable work will be needed to understand whether the observed defense activation pattern has any adaptive benefit.

Airborne information transfer has the potential to play a major role in plant communities, as it functions as a signaling viaduct between plants not physically connected to one another (*Karban, 2021*; *Hu et al., 2019*; *Wenig et al., 2019*). Although it has been established for decades that plants transmit airborne chemical information between individuals, we are only scratching the surface of the dynamic nature of this phenomenon. The kinetics of information transfer at the detailed temporal resolution of this study provide some insights regarding how HIPVs act across time and space. On this

**Table 2.** Aligned rank transformed nonparametric factorial repeated measures ANOVA results from data presented in *Figure 7*. Bold values: p<0.05 and underlined values: p<0.1. Abbreviations: HAC, (Z)-3-hexenyl acetate; SQT, sesquiterpenes; MNT, monoterpenes; DMNT, 4,8-dimethylnona-1,3,7-triene; TMTT, 4,8,12-trimethyltrideca-1,3,7,11-tetraene.

| | Herb | | HAC | | | Time | | | Herb × HAC | | | Herb × Time | | | HAC × Time | | | Herb × HAC × Time | | |
|---|---|---|---|---|---|---|---|---|---|---|---|---|---|---|---|---|---|---|---|---|
| | F | p | df | F | p | df | F | p | df | F | p | df | F | p | df | F | p | df | F | p |
| **SQT** | | | | | | | | | | | | | | | | | | | | |
| Base | 0.22 | 0.64 | 1,54 | 0.23 | 0.62 | NA | NA | NA | 1,54 | 0.05 | 0.82 | NA | NA | NA | NA | NA | NA | NA | NA | NA |
| D1 | 13.8 | **<0.001** | 1,54 | 0.40 | 0.53 | 4,216 | 8.03 | **<0.001** | 1,54 | 0.30 | 0.59 | 4,216 | 1.85 | 0.12 | 4,216 | 0.10 | 0.98 | 4,216 | 0.23 | 0.92 |
| Dark | 2.15 | 0.15 | 1,54 | 0.17 | 0.69 | 6,324 | 118 | **<0.001** | 1,54 | 0.28 | 0.60 | 6,324 | 1.65 | 0.13 | 6,324 | 1.06 | 0.39 | 6,324 | 0.83 | 0.55 |
| D2 | 8.39 | **0.005** | 1,54 | 8.69 | **0.005** | 5,270 | 12.8 | **<0.001** | 1,54 | 0.002 | 0.97 | 5,270 | 1.22 | 0.30 | 5,270 | 23.9 | **<0.001** | 2,570 | 0.93 | 0.46 |
| **MNT** | | | | | | | | | | | | | | | | | | | | |
| Base | 0.05 | 0.82 | 1,54 | 0.01 | 0.94 | NA | NA | NA | 1,54 | 1.33 | 0.25 | NA | NA | NA | NA | NA | NA | NA | NA | NA |
| D1 | 7.84 | **0.007** | 1,54 | 0.02 | 0.88 | 4,216 | 36.5 | **<0.001** | 1,54 | 2.03 | 0.16 | 4,216 | 1.17 | 0.32 | 4,216 | 0.47 | 0.76 | 4,216 | 0.83 | 0.51 |
| Dark | 2.75 | 0.10 | 1,54 | 1.41 | 0.24 | 6,324 | 28.1 | **<0.001** | 1,54 | 2.62 | 0.11 | 6,324 | 1.04 | 0.40 | 6,324 | 1.04 | 0.40 | 6,324 | 1.02 | 0.41 |
| D2 | 4.82 | **0.03** | 1,54 | 5.73 | **0.02** | 5,270 | 197 | **<0.001** | 1,54 | 1.68 | 0.20 | 5,270 | 2.57 | **0.03** | 5,270 | 9.90 | **<0.001** | 5,270 | 2.82 | **0.02** |
| **DMNT** | | | | | | | | | | | | | | | | | | | | |
| Base | 0.60 | 0.44 | 1,54 | 0.91 | 0.34 | NA | NA | NA | 1,54 | 3.52 | <u>0.07</u> | NA | NA | NA | NA | NA | NA | NA | NA | NA |
| D1 | 66.3 | **<0.001** | 1,54 | 2.55 | 0.12 | 4,216 | 81.4 | **<0.001** | 1,54 | 0.69 | 0.41 | 4,216 | 65.1 | **<0.001** | 4,216 | 0.62 | 0.65 | 4,216 | 1.10 | 0.36 |
| Dark | 2.34 | 0.13 | 1,54 | 0.45 | 0.50 | 6,324 | 10.7 | **<0.001** | 1,54 | 0.51 | 0.48 | 6,324 | 1.05 | 0.40 | 6,324 | 0.20 | 0.98 | 6,324 | 0.30 | 0.94 |
| D2 | 7.01 | **0.01** | 1,54 | 155 | **<0.001** | 5,270 | 52.4 | **<0.001** | 1,54 | 3.67 | <u>0.06</u> | 5,270 | 1.74 | 0.13 | 5,270 | 54.6 | **<0.001** | 5,270 | 2.09 | <u>0.07</u> |
| **TMTT** | | | | | | | | | | | | | | | | | | | | |
| Base | 0.12 | 0.73 | 1,54 | 0 | 0.99 | NA | NA | NA | 1,54 | 2.23 | 0.14 | NA | NA | NA | NA | NA | NA | NA | NA | NA |
| D1 | 8.91 | **0.004** | 1,54 | 0.67 | 0.42 | 1,54 | 3.05 | **0.02** | 1,54 | 0.01 | 0.92 | 4,216 | 1.43 | 0.23 | 4,216 | 0.58 | 0.68 | 4,216 | 0.43 | 0.79 |
| Dark | 2.68 | 0.11 | 1,54 | 0.33 | 0.57 | 1,54 | 7.67 | **<0.001** | 1,54 | 1.43 | 0.24 | 6,324 | 1.22 | 0.30 | 6,324 | 0.46 | 0.84 | 6,324 | 2.00 | <u>0.07</u> |
| D2 | 5.83 | **0.02** | 1,54 | 15.2 | **<0.001** | 1,54 | 5.28 | **<0.001** | 1,54 | 0.01 | 0.93 | 5,270 | 0.35 | 0.88 | 5,270 | 1.69 | 0.14 | 5,270 | 1.03 | 0.40 |

basis, a more comprehensive understanding of volatile-mediated information transfer between plants can be built.

## Materials and methods

### Plant and insect growth

V2-stage *Zea mays* (maize, B73) was used throughout this study. At this stage, maize has four leaves: two are fully developed, one is expanding and one is emerging. Maize plants were grown in commercial potting soil (Selmaterra, BiglerSamen, Switzerland) in 180 ml pots. Plants were grown in a greenhouse supplemented with artificial lights (ca. 300 µmol m$^{-2}$ s$^{-1}$). Between light and dark phases, plants were supplemented with lower light (ca. 60 µmol m$^{-2}$ s$^{-1}$) for 15 min to more accurately simulate day:night transitions. The greenhouse was maintained at 22 ± 2°C, 40–60% relative humidity, with a 14 hr:10 hr, light:dark cycle. *S. exigua* (Frontier Agricultural Sciences, USA) were reared from eggs on artificial diet (*Maag et al., 2014*) and used for experiments when they reached the fourth instar stage.

### Herbivore treatment and experimental setup

The herbivore treatment consisted of adding 3 fourth instar larvae onto plants (sender plants). For controls, sender plants were left undamaged. Two principal experimental setups were used to determine the kinetics of HIPV emissions in both sender plants (damaged by herbivores) and receiver plants (exposed to HIPVs from sender plants).

#### Setup 1

Single plants (senders) in transparent glass chambers (Ø×H 12×45 cm) were connected with PTFE tubing to a second transparent glass chamber that was either left empty or filled with a second plant (receiver). Chambers were sealed other than an airflow inlet on the first chamber and an air outlet on the second chamber, allowing HIPVs from sender plants to pass through the second chamber. Cumulative HIPV emissions from senders and receivers were measured as described in the section 'Volatile sampling'. In order to determine emissions from receiver plants alone, HIPV emissions from sender plants connected to an empty chamber were subtracted from sender plants connected to receiver plants (*Figure 8*).

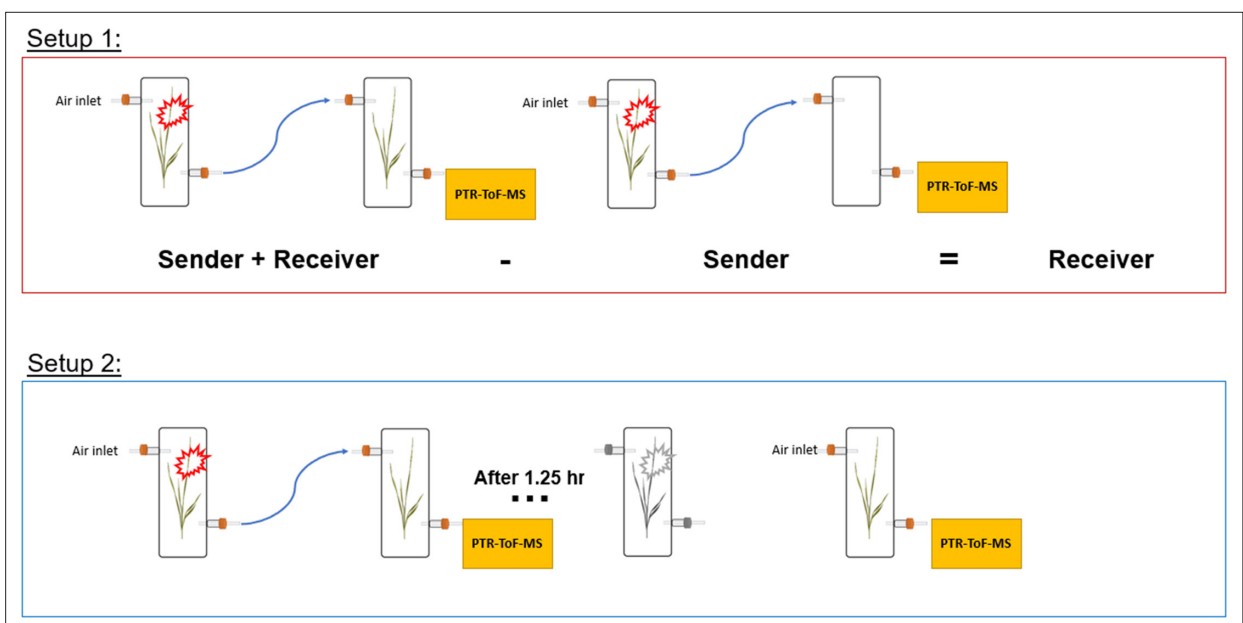

**Figure 8.** Experimental setup schemes for volatile profiling.

### Setup 2

To isolate emissions solely from receiver plants, chambers containing sender and receiver plants were initially kept separate for basal HIPV emission profiling. Chambers were then connected as in setup 1 for 30 min before herbivore treatment. After herbivores were added to sender plants, chambers remained connected for 1.25 hr, after which time sender and receiver plants were once again disconnected and rearranged so that receiver plants had both clean air flowing through the inlet and an outlet for volatile collection and profiling (*Figure 8*).

## Preparation of HAC dispensers

HAC dispensers were prepared as previously described with modifications (*Hu et al., 2019*; *Wang et al., 2023*). In brief, 1.5 ml glass vials (Ø×H 11.6×32 mm) containing ca. 100 mg glass wool were filled with 200 µl HAC (>98%; Sigma-Aldrich, Buchs, Switzerland) diluted 50-fold in EtOH and sealed with screw caps containing a rubber septum. The caps were pierced with a 1 µl glass capillary and sealed with PTFE tape. Vials were then covered with aluminum foil and equilibrated for at least 5 days before use. The (Z)-3 isomers of all GLVs measured in this study (hexenyl acetate, hexenal, and hexen-1-ol) are known to be bioactive (*Engelberth et al., 2004*; *Matsui et al., 2012*) and interconvertible, with hexenal and hexenyl acetate for instance being transformed into hexen-1-ol. As such, we designed the dispensers to emit a molar concentration of HAC comparable to the molar concentration of all (Z)-3 GLVs emitted by herbivore-infested plants. HAC dispensers emitted HAC at a rate of 4.68 (±0.40 SD) nmol hr$^{-1}$ and maize seedlings infested with 3 fourth instar *S. exigua* larvae emitted all GLVs at a rate of 1.94 nmol hr$^{-1}$ (±0.25 SD). Of note, GLV emissions induced by caterpillars vary over time, and can be more than twofold higher than the average during times of strong active feeding (*Figure 7—figure supplement 2*). Thus, the release rate of the dispensers is within the plant's physiological range.

## Volatile sampling

Volatile emissions were sampled using setups 1 and 2. Volatile profiling was measured with a high-throughput platform comprised of a PTR-ToF-MS (Tofwerk, Switzerland) and an automated headspace sampling system (Abon Life Sciences, Switzerland) supplied with clean airflow (0.8 l min$^{-1}$). An outlet on receiver plants was accessible to the autosampler/PTR-ToF-MS system. The PTR-ToF-MS system drew air at 0.1 l min$^{-1}$. Between samples, a zero gas measurement was performed for 3 s to avoid contamination. At each time point volatiles were continuously measured for 15–25 s and averaged to a single mean per sample. Complete mass spectra (0–500 m/z) were recorded in positive mode at ca. 10 Hz. The PTR was operated at 100°C and an E/N of approximately 120 Td. The volatile data extraction and processing were conducted using Tofware software package v3.2.2 (Tofwerk, Switzerland). Protonated compounds were identified based on their molecular weight+1. During volatile collection LED lights (DYNA, heliospectra) were placed ca. 80 cm above the glass cylinders and provided light at ca. 300 µmol m$^{-2}$ s$^{-1}$. Identical light:dark cycle timing as in the greenhouse for plant growth was used. In order to quantify GLVs emitted from herbivore-infested plants and HAC dispensers, we collected total volatile emissions for 1 hr using an identical volatile collection and analysis method described in *Erb et al., 2015*. In brief, volatiles were collected in Super-Q adsorbent traps, which were then eluted in dichloromethane and injected into a GC-MS (Agilent, USA). Compound identities were confirmed with mass spectroscopy analysis and similarity to library matches (NIST search 2.2 Mass Spectral Library, USA). Emission rates of the Z-3 isomers of GLVs were quantified using a standard curve with synthetic compounds.

## Foliar terpene pools and gene expression

Setup 1 was used to determine terpene pools and gene expression. After 3, 8, 16.75, and 22 hr of exposure to HIPVs, the oldest developing leaf of receiver plants was harvested and flash-frozen on liquid nitrogen. The oldest developing leaf was chosen as it is the largest and highly responsive to bioactive HIPVs (*Wang et al., 2023*). A new set of plants was used for each time point. Analysis of foliar terpene pools was conducted using slightly modified, previously described, methods (*Escobar-Bravo et al., 2022*). In brief, ca. 15 mg of ground fresh frozen leaf tissue was added to a 20 ml precision thread headspace glass vial sealed with a magnetic screw cap fitted with a silicone/PTFE septum (Gertel GmbH & Co. KG, Germany). Immediately after a vial was prepared, an SPME fiber (100 µm

polydimethylsiloxane coating; Supelco, USA) was inserted into the vial and volatiles were collected for 40 min at 50°C. After collection, volatiles were thermally desorbed for 3 min at 220°C and analyzed using GC-MS (Agilent, USA). Helium was used as the carrier gas at a flow-rate of 1 ml min$^{-1}$ with a temperature gradient of 5 °C min$^{-1}$ from 60°C (1 min hold) to 250°C. Compound identification was based on similarity to library matches (NIST search 2.2 Mass Spectral Library, USA). For quantification of gene expression, total RNA was extracted and purified from ca. 80 mg ground fresh frozen tissue using the GeneJET plant RNA extraction kit following the manufacturer's instructions. Genomic DNA was removed from 1 µg purified RNA using gDNA Eraser (PrimeScript RT Reagent Kit, Perfect Real Time) following the manufacturer's instructions (Takara Bio Inc, Kusatsu, Japan). Reverse transcription and cDNA was synthesized using PrimeScript Reverse Transcriptase (TaKaRa Bio). Gene expression was determined with quantitative reverse transcription polymerase chain reaction using ORA SEE qPCR Mix (highQu GmbH, Germany) on an Applied Biosystems QuantStudio 5 Real-Time PCR system. The normalized expression (NE) values were calculated as: $NE=(1/(PE_{target}{}^{Cttarget}))/(1/(PE_{reference}{}^{Ctreference}))$ where PE = primer efficiency and Ct = cycle threshold (*Alba et al., 2015*). Ubiquitin (UBI1) was used as the reference gene. Gene identifiers and primer sequences are listed in *Supplementary file 1*.

## Phytohormone quantification

Plant treatments and tissue collection for phytohormone were identical to foliar terpene and gene expression analysis. The phytohormones JA, JA-Ile, and OPDA were extracted, analyzed, and quantified using a slightly modified version of the method detailed in *Glauser et al., 2014*. In brief, ca. 80 mg of finely ground fresh frozen leaf tissue was extracted with 1 ml ethylacetate:formic acid (99.5:0.5, vol/vol) spiked with isotopically labeled forms of the abovementioned phytohormones. $d_5$-JA was acquired from CDN Isotopes (Canada), $d_5$-OPDA was acquired from OLChemIm (Czechia), and $^{13}C_6$-JA-Ile was synthesized at the University of Neuchâtel according to a previously described method (*Kramell et al., 1997*). Extracts were dried and re-suspended in 200 µl 50% MeOH using a sonicating bath for 30 min. Phytohormones were analyzed using UPLC-MS-MS fitted with an Acuity BEH C18 column (Waters, USA) using a flow-rate of 400 µl/min and an injection volume of 2 µl. The two mobile phases used were 0.05% formic acid in water (A) and 0.05% formic acid in acetonitrile (B) with the following gradient conditions: 5–50% B over 5 min, 60–100% B over 3 min, 100% B for 4 min, and a final re-equilibration at 5% B for 4 min.

## Statistical analyses

All statistical analyses were performed in R version 4.2.2 (*R Core Team, 2022*). Volatile emissions were analyzed by aligned rank transformed nonparametric factorial repeated measures ANOVA using the package ARTool, as individual plant emission kinetics were measured repeatedly over time (*Wobbrock et al., 2011*). Internal terpene pools, gene expression, and phytohormone levels were analyzed using Welch's *t*-tests between control and HIPV-exposed receiver plants within each time point.

## Acknowledgements

We would like to thank Dr. Bernardus Schimmel for assistance with gene expression quantification. Additionally, we thank all members of the Biotic Interactions and Chemical Ecology groups at the University of Bern for helpful discussions. This work was supported by the Swiss National Science Foundation (Grants Nr. 210651 and 200355), the State Secretariat for Education, Research, and Innovation SERI (Project CANWAS), the Horizon 2020 Marie Skłodowska-Curie Actions (Grant Nr. 886651), and the University of Bern.

## Additional information

### Funding

| Funder | Grant reference number | Author |
| --- | --- | --- |
| Schweizerischer Nationalfonds zur Förderung der Wissenschaftlichen Forschung | 210651 | Jamie Mitchel Waterman |
| Schweizerischer Nationalfonds zur Förderung der Wissenschaftlichen Forschung | 200355 | Matthias Erb |
| Staatssekretariat für Bildung, Forschung und Innovation | Project CANWAS | Matthias Erb |
| HORIZON EUROPE Marie Sklodowska-Curie Actions | 886651 | Lei Wang |
| University of Bern | | Jamie Mitchel Waterman Tristan Michael Cofer Lei Wang Matthias Erb |

The funders had no role in study design, data collection and interpretation, or the decision to submit the work for publication.

### Author contributions

Jamie Mitchel Waterman, Conceptualization, Data curation, Formal analysis, Funding acquisition, Validation, Investigation, Visualization, Methodology, Writing – original draft, Project administration, Writing – review and editing; Tristan Michael Cofer, Resources, Data curation, Formal analysis, Investigation, Methodology, Writing – review and editing; Lei Wang, Resources, Funding acquisition, Methodology, Writing – review and editing; Gaetan Glauser, Resources, Formal analysis, Methodology; Matthias Erb, Conceptualization, Resources, Supervision, Funding acquisition, Methodology, Project administration, Writing – review and editing

### Author ORCIDs

Jamie Mitchel Waterman ⓘ https://orcid.org/0000-0003-1782-827X
Tristan Michael Cofer ⓘ http://orcid.org/0000-0001-6175-9668
Lei Wang ⓘ http://orcid.org/0000-0002-6332-6476
Gaetan Glauser ⓘ http://orcid.org/0000-0002-0983-8614
Matthias Erb ⓘ http://orcid.org/0000-0002-4446-9834

Reviewer #1 (Public Review): https://doi.org/10.7554/eLife.89855.3.sa1
Reviewer #2 (Public Review): https://doi.org/10.7554/eLife.89855.3.sa2
Author Response https://doi.org/10.7554/eLife.89855.3.sa3

## Additional files

### Supplementary files

• Supplementary file 1. Primers used in this study.

• MDAR checklist

• Source data 1. Source data used to produce all figures and tables included in the main and supplemental text.

## Data availability

Data used to produce all results presented in the main text and supplemental material are attached as *Source data 1*.

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
