## [Editor Report · eLife assessment]

This **fundamental** study examines the effects of herbivory-induced maize volatiles on neighbouring plants and their responses over time. Measurements of volatile compound classes and gene expression in receiver plants exposed to these volatiles led to the conclusion that the delayed emission of certain terpenes in receiver plants after the onset of light may be a result of stress memory, highlighting the role of priming and induction in plant defences triggered by herbivore-induced plant volatiles. The evidence supporting the conclusions is **compelling**, with rigorous chemical assays of and state-of-the-art high throughput real time mass spectrometry. The work will be of broad interest to plant biologists and chemical ecologists.

---

## [Referee Report · Reviewer #1 (Public Review)]

The authors of the manuscript "High-resolution kinetics of herbivore-induced plant volatile transfer reveal tightly clocked responses in neighboring plants" assessed the effects of herbivory induced maize volatiles on receiver plants over a period of time in order to assess the dynamics of the responses of receiver plants. Different volatile compound classes were measured over a period of time using PTR-ToF-MS and GC-MS, under both natural light:dark conditions, and continuous light. They also measured gene expression of related genes as well as defense related phytohormones. The effects of a secondary exposure to GLVs on primed receiver plants was also measured.

The paper addresses some interesting points, however some questions arise regarding some of the methods employed. Firstly, I am wondering why VOCs (as measured by GC-MS) were not quantified. While I understand that quantification is time consuming and requires more work, it allows for comparisons to be made between lines of the same species, as well as across other literature on the subject. Simply relying on the area under the curve and presenting results using arbitrary units is not enough for analyses like these. AU values do not allow for conclusions regarding total quantities, and while I understand that this is not the main focus of this paper, it raises a lot of uncertainty for readers (for example, the references cited show that TMTT has been found to accumulate at similar levels of caryophyllene, however the AU values reported are an order of magnitude higher for TMTT. Again, without actual quantification this is meaningless, but for readers it is confusing).

With regards to the correlation analyses shown in figure 6, the results presented in many of the correlation plots are not actually informative. While there is a trend, I do not think that this is an appropriate way to show the data, as there are clearly other relationships at play. The comparison between plants under continuous light and normal light:dark conditions is interesting.

This paper addresses a very interesting idea and I look forward to seeing further work that builds on these ideas.

---

## [Referee Report · Reviewer #2 (Public Review)]

The exact dynamics of responses to volatiles from herbivore-attacked neighbouring plants have been little studied so far. Also, we still lack evidence whether herbivore-induced plant volatiles (HIPVs) induce or prime plant defences of neighbours. The authors investigated the volatile emission patterns of receiver plants that respond to the volatile emission of neighbouring sender plants which are fed upon by herbivorous caterpillars. They applied a very elegant approach (more rigorous than the current state-of-the-art) to monitor temporal response patterns of neighbouring plants to HIPVs by measuring volatile emissions of senders and receivers, senders only and receivers only. Different terpenoids were produced within 2 h of such exposure in receiver plants, but not during the dark phase. Once the light turned on again, large amounts of terpenoids were released from the receiver plants. This may indicate a delayed terpene burst, but terpenoids may also be induced by the sudden change in light. As one contrasting control, the authors also studied the time-delay in volatile emission when plants were just kept under continuous light. Here they also found a delayed terpenoid production, but this seemed to be lower compared to the plants exposed to the day-night-cycle. Another helpful control was now performed for the revision in which the herbivory treatment was started in the evening hours and lights were left on. This experiment revealed that the burst of terpenoid emission indeed shifted somewhat. Circadiane and diurnal processes must thus interact.

Interestingly, internal terpene pools of one of the leaves tested here remained more comparable between night and day, indicating that their pools stay higher in plants exposed to HIPVs. In contrast, terpene synthases were only induced during the light-phase, not in the dark-phase. Moreover, jasmonates were only significantly induced 22 h after onset of the volatile exposure and thus parallel with the burst of terpene release.

An additional experiment exposing plants to the green leaf volatile (glv) (Z)-3-hexenyl acetate revealed that plants can be primed by this glv, leading to a stronger terpene burst. The results are discussed with nice logic and considering potential ecological consequences. All data are now well discussed.

Overall, this study provides intriguing insights in the potential interplay between priming and induction, which may co-occur, enhancing (indirect and direct) plant defence. Follow-up studies are suggested that may provide additional evidence.

---

## [Author Response]

The following is the authors’ response to the current reviews.

**Reviewer #1 (Public Review):**
The authors of the manuscript "High-resolution kinetics of herbivore-induced plant volatile transfer reveal tightly clocked responses in neighboring plants" assessed the effects of herbivory induced maize volatiles on receiver plants over a period of time in order to assess the dynamics of the responses of receiver plants. Different volatile compound classes were measured over a period of time using PTR-ToF-MS and GC-MS, under both natural light:dark conditions, and continuous light. They also measured gene expression of related genes as well as defense related phytohormones. The effects of a secondary exposure to GLVs on primed receiver plants was also measured.The paper addresses some interesting points, however some questions arise regarding some of the methods employed. Firstly, I am wondering why VOCs (as measured by GC-MS) were not quantified. While I understand that quantification is time consuming and requires more work, it allows for comparisons to be made between lines of the same species, as well as across other literature on the subject. Simply relying on the area under the curve and presenting results using arbitrary units is not enough for analyses like these. AU values do not allow for conclusions regarding total quantities, and while I understand that this is not the main focus of this paper, it raises a lot of uncertainty for readers (for example, the references cited show that TMTT has been found to accumulate at similar levels of caryophyllene, however the AU values reported are an order of magnitude higher for TMTT. Again, without actual quantification this is meaningless, but for readers it is confusing).With regards to the correlation analyses shown in figure 6, the results presented in many of the correlation plots are not actually informative. While there is a trend, I do not think that this is an appropriate way to show the data, as there are clearly other relationships at play. The comparison between plants under continuous light and normal light:dark conditions is interesting.This paper addresses a very interesting idea and I look forward to seeing further work that builds on these ideas.

As mentioned in our previous response, we have added the quantification of GLVs in order to increase the comparability of our work to other studies.

Regarding the comment about TMTT (only measured as internal pools), the purpose of the inclusion of these internal pool data, was simply to determine whether terpenes were accumulating in leaf tissue during the night when emissions are hindered (likely due to closed stomata). The data clearly show that internal terpene pools do not accumulate above daytime levels during darkness – this is further supported by gene expression data that show downregulation of terpene synthase genes during darkness. While quantification would certainly increase the ability to compare internal pools, it would not change the interpretation of our results. Also note that absolute quantification is challenging for compounds such as TMTT, which are not readily available.

Regarding the comment on Figure 6, while we agree there may be interesting patterns beyond linear relationships, as stated in our previous response, the purpose of our analysis was to determine if the higher terpene burst in receiver plants on the second day may be explained by sender plants emitting more GLVs on the second day. Figure 6 shows that this is not the case. Further analyses would not provide additional significant insights into the hypothesis that we tested here.

We thank the reviewer for their overall positive outlook on our paper and for the constructive comments.

**Reviewer #2 (Public Review):**
The exact dynamics of responses to volatiles from herbivore-attacked neighbouring plants have been little studied so far. Also, we still lack evidence whether herbivore-induced plant volatiles (HIPVs) induce or prime plant defences of neighbours. The authors investigated the volatile emission patterns of receiver plants that respond to the volatile emission of neighbouring sender plants which are fed upon by herbivorous caterpillars. They applied a very elegant approach (more rigorous than the current state-of-the-art) to monitor temporal response patterns of neighbouring plants to HIPVs by measuring volatile emissions of senders and receivers, senders only and receivers only. Different terpenoids were produced within 2 h of such exposure in receiver plants, but not during the dark phase. Once the light turned on again, large amounts of terpenoids were released from the receiver plants. This may indicate a delayed terpene burst, but terpenoids may also be induced by the sudden change in light. As one contrasting control, the authors also studied the time-delay in volatile emission when plants were just kept under continuous light. Here they also found a delayed terpenoid production, but this seemed to be lower compared to the plants exposed to the day-night-cycle. Another helpful control was now performed for the revision in which the herbivory treatment was started in the evening hours and lights were left on. This experiment revealed that the burst of terpenoid emission indeed shifted somewhat. Circadiane and diurnal processes must thus interact.Interestingly, internal terpene pools of one of the leaves tested here remained more comparable between night and day, indicating that their pools stay higher in plants exposed to HIPVs. In contrast, terpene synthases were only induced during the light-phase, not in the dark-phase. Moreover, jasmonates were only significantly induced 22 h after onset of the volatile exposure and thus parallel with the burst of terpene release.An additional experiment exposing plants to the green leaf volatile (glv) (Z)-3-hexenyl acetate revealed that plants can be primed by this glv, leading to a stronger terpene burst. The results are discussed with nice logic and considering potential ecological consequences. All data are now well discussed.Overall, this study provides intriguing insights in the potential interplay between priming and induction, which may co-occur, enhancing (indirect and direct) plant defence. Follow-up studies are suggested that may provide additional evidence.

We thank the reviewer for their positive outlook on our paper and for their constructive comments.

**Recommendations for the authors:**

**Reviewer #2 (Recommendations For The Authors):**
The authors did a great job with the revision. The additional experiments strengthened their conclusions. Thanks also for performing the suggested test for potential differences in induction capacity at different times of day, the new data are very interesting.

Thank you very much.

Line 49-52: The newly added sentence could be clarified in wording.

We will clarify the sentence.

Line 254-255: The newly added sentence needs to be corrected. This is no full sentence and it is not clear what the authors wanted to say here.

We will clarify this sentence.

Figure 6: In those instances, in which the correlation is not significant, the line should not be shown.

We will remove the lines when correlations are not significant.

The names of chemical compounds and terpene synthases should be written in lower case letters (see legend Fig 6, e.g. hexenal, not Hexenal; legend fig. 2: terpene synthase, not Terpene synthase)In the last round of revisions, I commented on Line 23: consequences on community dynamics are not investigated here, so this is a bit misleading. ... Your response was "We have deleted the sentence about community dynamics ..." which, however, in fact was not done! Please change!

Apologies for that, we will delete mention of community dynamics in that sentence (for real).

The following is the authors’ response to the original reviews.

**eLife assessment**
This important study examines the effects of herbivory-induced maize volatiles on neighboring plants and their responses over time. Measurements of volatile compound classes and gene expression in receiver plants exposed to these volatiles led to the conclusion that the delayed emission of certain terpenes in receiver plants after the onset of light may be a result of stress memory, highlighting the role of priming and induction in plant defenses triggered by herbivore-induced plant volatiles (HIPVs). Most experimental data are compelling but additional experiments and accurate quantifications of the compounds would be required to confirm some of the main claims.

Response: We thank the editors for their overall positive feedback on our MS. We have added additional experiments to quantify green leaf volatile emissions in both sender plants and synthetic dispensers (Reviewer 1) and address the importance of the precise time of day plants are induced (Reviewer 2). These additions strengthen the main conclusions of our study.

**Public Reviews:**

**Reviewer #1 (Public Review):**
The authors of the manuscript "High-resolution kinetics of herbivore-induced plant volatile transfer reveal tightly clocked responses in neighboring plants" assessed the effects of herbivory-induced maize volatiles on receiver plants over a period of time in order to assess the dynamics of the responses of receiver plants. Different volatile compound classes were measured over a period of time using PTR-ToF-MS and GC-MS, under both natural light:dark conditions, and continuous light. They also measured gene expression of related genes as well as defence-related phytohormones. The effects of a secondary exposure to GLVs on primed receiver plants were also measured.The paper addresses some interesting points, however, some questions arise regarding some of the methods employed. Firstly, I am wondering why VOCs (as measured by GC-MS) were not quantified. While I understand that quantification is time-consuming and requires more work, it allows for comparisons to be made between lines of the same species, as well as across other literature on the subject. As experiments with VOC dispensers were also used in this experiment, I find it even more baffling that the authors didn't confirm the concentration of the emission from the plants they used to make sure they matched. The references cited justifying the concentration used (saying it was within the range of GLVs emitted by their plants) to prepare the dispenser were for either a different variety of maize (delprim versus B73) or arabidopsis. Simply relying on the area under the curve and presenting results using arbitrary units is not enough for analyses like these.

Response: We thank the reviewer for their comment. We have now quantified both the emission of dispensers and maize seedlings infested with 3 4th-instar Spodoptera exigua larvae. Averaged across 1 h, HAC dispensers emitted roughly 2x higher molar concentrations than total GLV molar concentrations emitted by plants infested by 3 caterpillars. Of note, GLV emissions induced by caterpillars vary over time, and can be more than 2-fold higher than the average during times of strong active feeding (Supplemental Fig 4). Thus, the release rate of the dispensers is well within the plant’s physiological range.

Note that the references cited were included to support the claim of the biological activity of all three GLVs rather than to justify concentration of our dispensers. We have rephrased this sentence to reflect this (see L330-333).

With regards to the correlation analyses shown in Figure 6, the results presented in many of the correlation plots are not actually informative. By blindly reporting the correlation coefficient important trends are being ignored, as there are clearly either bimodal relationships (e.g. upper left panel, HAC/TMTT, HAC/MNT) or even stranger relationships (e.g. upper left panel, IND/SQT, IND/MNT) that are not being well explained by a correlation plot. It is not appropriate to discuss the correlation factors presented here and to draw such strong conclusions on emission kinetics. The comparison between plants under continuous light and normal light:dark conditions is interesting, but I think there are better ways to examine these relationships, for example, multivariate analysis might reveal some patterns.

Response: We thank the reviewer for their comment. With our analysis we aimed at testing specifically whether the high release of known bioactive volatiles (GLVs and indole) by sender plants on the second day can explain the higher terpene emissions in the receiver plants. We explicitly mention this in the text (L176-L186). Indeed, under normal light conditions (light and dark phase), there are clear positive correlations between the GLV release of sender plants and the terpene release of receiver plants over time (see also Fig 1 and Fig 5). However, under continuous light conditions, GLV emissions in sender plants no longer correlate with terpene emissions in receiver plants (also apparent by comparison of Fig 4 and Fig 5). This shows that temporal variation in GLV emissions are insufficient to explain the delayed terpene burst. This is the relevant conclusion we draw from this analysis. As presented, we find the data to provide strong evidence that the delayed burst in receiver plant terpene emissions cannot be solely explained by higher availability of active signals on the second day. The priming experiment in Figure 7 then provides a direct additional test for this concept. While more complex analyses could indeed reveal additional patterns, these would not be particularly informative for the question at hand.

In Figure 2, the elevated concentrations of beta-caryophyllene found in the control plants at 8h and 16.75h measurement timepoints are curious. Is this something that is commonly seen in B73?

Response: We thank the reviewer for this comment. A small number of untreated plants indeed accumulated β -caryophyllene at night, which is likely the result of biological variability between samples. Our plants were soil-grown, and it is for instance possible that variation in soil biota may account for this variability. Alternatively, some plants may have been slightly stressed during handling. Note that this variability does not affect any of the conclusions in our manuscript.

While there can be discrepancies between emissions and compounds actually present within leaf tissue, it is a little bit odd that such high levels of b-caryophyllene were found at these timepoints, however, this is not reflected in the PTR-ToF-MS measurements of sesquiterpenes. It would be beneficial to include an overview of the mechanism of production and storage of sesquiterpenes in maize leaves, which would clarify why high amounts were found only in the GC-MS analysis and not the PTR-ToF-MS analysis, which is a more sensitive analytical tool. It is possible that the amounts of b-caryophyllene present in the leaf are actually extremely low, however as the values are not given as a concentration but rather arbitrary units, it is not possible to tell. I would include a line explaining what is seen with b-caryophyllene.

Response: Thank you for this comment. It is important to note that accumulation in maize leaves can differ substantially from emission, especially at night when stomata are closed. This has been observed before in maize leaves (Seidl-Adams et al., 2015). As the reviewer suspects, earlier work indeed found that β-caryophyllene is a minor sesquiterpene compared to β-farnesene and α-bergamotene in B73 ( Block et al., 2018). The PTR-ToF-MS does not discriminate between terpenes with the same m/z and thus measures total sesquiterpene emissions. Given that sesquiterpene emissions are strongly regulated by stomatal aperture and that overall sesquiterpene accumulation in control plants is low, it is not surprising that we measure only minor amounts of sesquiterpene emissions in general, and in control plants in particular. We now text to the manuscript to explain these aspects (L116-L122).Block, A.K., Hunter, C.T., Rering, C. et al. Contrasting insect attraction and herbivore-induced plant volatile production in maize. Planta 248, 105–116 (2018).

Seidl-Adams I, Richter A, Boomer KB, Yoshinaga N, Degenhardt J, Tumlinson JH. Emission of herbivore elicitor-induced sesquiterpenes is regulated by stomatal aperture in maize (Zea mays) seedlings. Plant Cell Environ. 38, 23-34 (2015).

Additionally, it seems like the amounts of TMTT within the leaf are extraordinarily high (judging only by the au values given for scale), far higher than one would expect from maize.

Response: We are unsure about the reviewer’s interpretation here. The AU values do not allow for conclusions regarding total quantities. An earlier study found that TMTT in induced B73 plants accumulates to similar amounts as β-caryophyllene (Block et al., 2018), thus it is not surprising to detect significant TMTT pools in induced maize leaves. It is important to note that the aim of the experiment here was to test the hypothesis that plants may be hyperaccumulating volatiles when the stomata are closed at night, which could potentially explain the delayed terpene burst on the second day. We do not observe such a hyperaccumulation, thus ruling out this as the primary factor responsible for the observed phenomenon. This is further supported by the continuous light experiments, where the delayed burst in terpene emission is not hindered by the lack of a dark phase.

Block, A.K., Hunter, C.T., Rering, C. et al. Contrasting insect attraction and herbivore-induced plant volatile production in maize. Planta 248, 105–116 (2018).

**Reviewer #2 (Public Review):**
The exact dynamics of responses to volatiles from herbivore-attacked neighbouring plants have been little studied so far. Also, we still lack evidence of whether herbivore-induced plant volatiles (HIPVs) induce or prime plant defences of neighbours. The authors investigated the volatile emission patterns of receiver plants that respond to the volatile emission of neighbouring sender plants which are fed upon by herbivorous caterpillars. They applied a very elegant approach (more rigorous than the current state-of-the-art) to monitor temporal response patterns of neighbouring plants to HIPVs by measuring volatile emissions of senders and receivers, senders only and receivers only. Different terpenoids were produced within 2 h of such exposure in receiver plants, but not during the dark phase. Once the light turned on again, large amounts of terpenoids were released from the receiver plants. This may indicate a delayed terpene burst, but terpenoids may also be induced by the sudden change in light. A potential caveat exists with respect to the exact timing and the day-night cycle. The timing may be critical, i.e. at which time-point after onset of light herbivores were placed on the plants and how long the terpene emission lasted before the light was turned off. If the rhythm or a potential internal clock matters, then this information should also be highly relevant. Moreover, light on/off is a rather arbitrary treatment that is practical for experiments in the laboratory but which is not a very realistic setting. Particularly with regard to terpene emission, the sudden turning on of light instead of a smooth and continuous change to lighter conditions may trigger emission responses that are not found in nature.

Response: We thank the reviewer for their comment. Although not explicitly mentioned it in the initial draft of the MS, we employed 15 min transition periods for light and dark phase transitions with a light intensity of 60 µmol m-2 s-1 (compared to 300 µmol m-2 s-1 at full light) to achieve a more gradual transition. We now included this information in the manuscript (L291-L292).

As one contrasting control, the authors also studied the time-delay in volatile emission when plants were just kept under continuous light (just for the experiment or continuously?). Here they also found a delayed terpenoid production, but this seemed to be lower compared to the plants exposed to the day-night-cycle. Another helpful control would be to start the herbivory treatment in the evening hours and leave the light on. If then again plants only release volatiles after a 17 h delay, the response is indeed independent of the diurnal clock of the plant.

Response: This is a very interesting point raised by the reviewer. We now conducted an additional experiment under continuous light where we started the herbivory treatment just before the start of the dark phase (ca. 20:00 PM). We found a similar pattern: a distinct delay in the highest burst. However, interestingly, the burst was shifted from 12-18 hr to 10-12 hr (Supplemental Fig 1). This burst aligned reasonably well with the point at which lights would normally be turned on again. In light of this, and, as the herbivore additions typically started ca. 5 hrs after the onset of light following a dark period (Figures 1-7), we wanted to rule out the possibility that the lack of a burst on the first day, was simply due to a difference in induction capacity depending on how shortly after the onset of light plants became exposed to GLVs. As such, we designed an additional experiment to examine whether exposure to GLVs immediately after the lights come on induce higher terpene emissions than plants exposed to GLVs ca. 5 hr after lights come on (Supplemental Fig 2). Interestingly, emissions across the terpenes were similar, regardless how long after the onset of lights on plants were exposed to GLVs. This suggests that the delayed burst is not due to the fact that, on the second day, plants are exposed to GLVs immediately after the lights come on whereas the first day they are only exposed 5 hr after the lights come on. Both continuous light experiments (normal timing and shifted timing) show bursts that occur slightly earlier than we observe with under normal day : night light conditions (L159-L166 and L207-L211), suggesting an interaction between circadian and diurnal processes. For instance, it is possible that plants would start producing volatiles slightly earlier than the onset of the day, however, light and stomatal opening limits the exact timing of the burst under normal light:dark transitions. The additional data provide further evidence for the delayed burst as a timed response in maize plants.

Additionally, we have added explanation the continuous light figure legends that plants were grown under normal conditions and lights were only left on following treatment.

Interestingly, internal terpene pools of one of the leaves tested here remained more comparable between night and day, indicating that their pools stay higher in plants exposed to HIPVs. In contrast, terpene synthases were only induced during the light-phase, not in the dark-phase. Moreover, jasmonates were only significantly induced 22 h after the onset of the volatile exposure and thus parallel with the burst of terpene release. An additional experiment exposing plants to the green leaf volatile (glv) (Z)-3-hexenyl acetate revealed that plants can be primed by this glv, leading to a stronger terpene burst. The results are discussed with nice logic and considering potential ecological consequences. Some data are not discussed, e.g. the jasmonate and gene induction pattern.

Response: Thanks for this comment. We have added a sentence regarding the jasmonate data suggesting that, in addition to providing an additional layer of evidence for the observed delay, suggest that other JA-dependent defenses in maize may follow similar temporal patterns (L254-L257).

Overall, this study provides intriguing insights into the potential interplay between priming and induction, which may co-occur, enhancing (indirect and direct) plant defence. Follow-up studies are suggested that may provide additional evidence.
**Reviewer #1 (Recommendations For The Authors):**
Could the authors please explain why they chose not to calculate concentrations for VOCs? Perhaps it is that B73 is a very unique variety in that it contains very high levels of TMTT, even in control plants? This should be clarified by the authors.

Response: We address this comment in the public review portion

For the legend within Figure 2, I would move it to be in the upper left or right corners of the figure. It is not easy to see in its current position.

Response: We have moved the figure legend based on the reviewers recommendation

Figures depicting PTR-ToF-MS data: add m/z values to either the figures themselves and/or the legends.

Response: We have added m/z values to the legends and added molecular formulas of protonated compounds to each panel.

Overall, here are some other suggestions: I am slightly weary of the term "clocked response". I'm not sure this is the correct fit for what you are trying to convey. I think "regulated" is a better term than "clocked". I understand that it is likely a stylistic choice to use this word, however, I advise reconsidering for the sake of clarity of the results.

Response: Thank you. We find clocked to be an appropriate term, as it highlights the temporal aspect of the burst, and have thus left the title as is.

Have another look at the references as some are not in the correct format (i.e., species not in italics).

Response: We have checked and corrected the references

**Reviewer #2 (Recommendations For The Authors):**
Line 23: consequences on community dynamics are not investigated here, so this is a bit misleading.Last sentence of the abstract: It would be nice to read the answer to this long-standing question here.

Response: We have deleted he sentence about community dynamics and provided a more concrete final sentence (L38-L40)

Lines 48-50: The example does not fit so well with the first sentence and is not entirely clear (relation to temporal dynamics; similar to what?).

Response: We have reworded the sentence for clarity (L49-L52)

Line 56: "volatiles" should be plural.

Response: Changed (L58)

Line 58: "to be produced" rather than "to produce"

Response: This seems a stylistic choice, and have left it as is.

End of abstract: Did you have any hypotheses? These should be stated here.

Response: The listing of hypotheses is also a stylistic choice, which is in some cases required by journals, but not eLife. As such we have not included a discrete list of hypotheses and instead describe what we aimed to investigate and what we found.

Line 93: "This response disappeared at night." Does this mean: "No volatiles were emitted during night"? Or was this a gradual disappearance? How many hours after the onset of light did the herbivore treatment start and how many hours after the first emission of volatiles was the light turned off?

Response: We have added when herbivory began (L92-L93) and changed the text to ‘as soon as light was restored’ (L97-L98).

Line 93: "as soon as the night was over" means practically rather "as soon as the light was switched on".

Response: See above

Line 91: "small induction" - do you mean "low amounts of xxx"?

Response: We mean a small induction. Terpene emission is relatively low (hence small), but still induced relative controls.

Line 91: which mono- and sesquiterpenes were monitored?

Response: It is PTR-ToF-MS a thus we cannot identify individual sesquiterpenes and monoterpenes (as they all have the same mass), and thus group them generally.

Figure 1: What exactly is the "control"? And what does the vertical hatched line in the beginning represent?

Response: We have defined the control and added a sentence describing the vertical hatched line

"Black points represent the same but with undamaged sender plants" - what is "the same" here? I find that a bit confusing!

Response: We have rephrased

Line 104: how do you define an "overaccumulation"?

Response: We have added ‘above daytime levels’ to clarify that we mean over daytime levels (L106)

Why was the oldest developing leaf chosen? Is this the largest one when plants are two weeks old? How many leaves do they have then? Is this the leaf with the highest biomass?

Response: We chose this leaf as it is the largest and also highly responsive to HIPVs. We have added this sentence (with a reference) in the methods section (L369-L370)

Line 107: "started increasing after 3 hours" - they may already have started before. The following description also sounds like the dynamics were investigated here. However, instead the authors measured samples at four distinct time-points and cannot say whether something "began" or "remained" etc. The wording should be changed to a more appropriate description, describing the differences at a given time-point.

Response: We changed the wording to ‘were marginally induced after 3 hr’ see L110

Line 113: What do you mean by "delete BELOW NIGHTTIME levels"?

Response: The word we used was ‘deplete’ to ‘drop’ (L116)

Line 114: "the expression of terpene synthases" add "in the receiver plants exposed to HIPVs."

Response: Added

Figure 2ff: The situation of receiver plants exposed to control plant volatiles is not explained in the method section and also not depicted in the Suppl. Fig. 1. Here, the sender plants seem to always have been induced (if the red star-like structure should resemble an induction - a legend may be helpful here).

Response: We have changed to ‘connected to undamaged sender plants’. We additionally added a sentence to the methods section describing controls L300

Line 140: This treatment is not described in the methods section. Were the plants only kept under constant conditions for the 2 experimental days? Compared to the induction shown in Fig. 1, the amount of released volatiles seems less here.

Response: We have added explanation of this to the figure legends, explaining that plants were grown under normal conditions and lights were only left on following treatment

Another helpful control would be to start the herbivory treatment in the evening hours and leave the light on. If then again plants only release volatiles after a 17 h delay, the response is indeed independent of the diurnal clock of the plant.

Response: See public review comment. We have added this experiment and discuss it accordingly in the MS (L159-L166 and L207-L211)

Line 157: Check sentence/grammar

Response: Checked and modified

Figure 5: I suggest using a different colour for volatiles released from the sender plants, not again the green also used in the other figures for the receiver plants. This would help the reader to quickly see which plants are in focus in each figure.

Response: We have changed the color of the figures for clarity

Figure 6 legend: check grammar in several sentences (use of singular vs. plural)

Response: We have made the tense uniform

The diurnal rhythm of jasmonates (and potentially also terpene synthases?) is not considered in the discussion.

Response: See above, and we have added a sentence to the discussion mentioning the jasmonates (L254-L257)

Line 230-231: check grammar. Given the complexity, the response pattern may not be so predictable.

Response: We do not understand this comment, but have checked the grammar throughout the manuscript.

Line 235: I like the discussion on potential ecological consequences.While some interpretation for each experiment is already given in the results section, not all results are discussed in the discussion section. For example, the jasmonate data are not discussed. This should be added.

Response: See above

Line 266: To get an idea about the plant size: How many leaves do the plants have in that stage?

Response: Added a sentence describing the size L287-L288

Line 321: change to "as in the greenhouse"

Response: Changed

Line 334: How were the terpenoids identified and, in particular, quantified?

Response: Added (L379-L380)

Line 354: Maybe rather change to: "Plant treatments and tissue collection for phytohormone sampling were identical as described above for terpene and gene expression analysis.

Response: Changed

Line 357: add "material" or "leaf tissue" after "flash frozen"

Response: Added

Line 359: What was the source of the isotopically labelled phytohormones?

Response: Added (L400-L403)

Line 360: The phytohormones are "analyzed" using UPLC. The "quantification" is then done afterward. Please correct.

Response: Corrected (L404)

Overall: a great approach and a wonderful idea!

Thanks